# Structural expression of a fading rift front, a case study from the Oligo-Miocene Irbid rift of northwest Arabia

Reli Wald[1], Amit Segev[2], Zvi Ben-Avraham[3], Uri Schattner[1]

[1]The Dr. Moses Strauss Department of Marine Geosciences, Leon H. Charney School of marine sciences, Haifa University,
Mt. Carmel, Haifa 31905, Israel.
[2]Geological Survey of Israel, 30 Malkhe Israel, Jerusalem 95501, Israel.
[3]Department of Geophysics and Planetary Sciences, Tel Aviv University, Tel Aviv 69978, Israel.

*Correspondence to*: Reli Wald (rwald@campus.haifa.ac.il)

**Abstract.**

Not all continental rifts mature to form a young ocean. The mechanism and duration of their cessation depend on the crustal structure, modifications in plate kinematics, lithospheric thermal response, or intensity of sub-crustal flow (e.g., plume activity). The cessation is recorded in the structure and stratigraphy of the basins that develop during the rifting process. This architecture is lost due to younger tectonic inversion, severe erosion or even burial into greater depths that forces their detection

by low-resolution geophysical imaging. The current study focuses on a uniquely preserved Oligo-Miocene rift that was subsequently taken over by a crossing transform fault system and, mostly due to that, died out. We integrate all geological, geophysical and results from previous studies from across the Southern Galilee to unravel the structural development of the Irbid failing rift, of Northwest Arabia. Despite tectonic, magmatic and geomorphologic activity postdating the rifting, its subsurface structure northwest of the Dead Sea Fault is preserved at depths of up to 1 km. Our results show that a series of

basins subsided at the rift front, i.e. rift termination, across the southern Galilee. We constrain the timing and extent of their subsidence into two main stages, based on facies analysis and chronology of magmatism. Between 20-9 Ma grabens and half-grabens subsided within a larger releasing jog, following an NW direction of a deeper presumed Principal Displacement Zone. The basins continued to subside until a transition from the transtenssional Red Sea to the transpressional Dead Sea stress regime occurred. With the transition, the basins ceased to subside as a rift, while the Dead Sea Fault split the jog structure.

Between 9-5 Ma basin subsidence accentuated and an uplift of their margins accompanied their overall elongation to the NNE. Our study provides for the first time a structural as well as tectonic context for the southern Galilee basins. Based on this case study we suggest that the rift did not fail but rather faded and was taken over by a more dominant stress regime. Otherwise, these basins of a failing rift could have simply died out peacefully.

## 1. Introduction

Failed continental rifts mark regions where crustal extension began in the past but did not mature into continental breakup. Their extension first forms an elongated valley that hosts a series of subsiding basins. Seismicity and volcanism accompany the subsidence, as observed along the Rhine Graben, the East African Rift, the Baikal Rift and the Shanxi Rift of China (Ziegler

and Cloetingh, 2004). However, some rifts fail to mature beyond this stage. Their seismicity, volcanism, and overall extension gradually cease. They become aulacogens, also called failed-, palaeo-, and aborted-rifts (Hoffman et al., 1974; Şengör, 1995; Brueseke et al., 2016).

Rifting cessation may result from modifications in plate kinematics, or in lithospheric thermal re-equilibration (e.g., along the Ordovician-Silurian Transbrasiliano lineament; Oliveira and Mohriak, 2003). It could also reflect a decay in plume intensity (e.g, Delhi basin; Sharma, 2009) or variations in rheological properties (Lyakhovsky et al., 2012). In this case, the extensional strain is accommodated by localized deformations over a wider region than the original rift axis (Van Wijk and Blackman, 2005, Segev et al., 2014).

The mechanism and duration of the cessation vary from one case to another. A rapid stop might be a result of extensional stress decay, acquaintance with a more rigid crust, or a newly established stress regime, different enough to mute the rifting process. Fading, i.e. gradual decrease of the dominant rifting stress leads to attenuation and eventually rift abortion. In the Potiguar rift (Brazil) case, Precambrian basement faulting patterns dictated the Neocomian-Barremian syn-rift graben formation style. Magnetic, gravity and resistivity data delineated intraplate transform boundaries, which generated fault-controlled depressions. Both the NE-trending (parallel to rift axis) oblique-slip faults and the NS-trending en-echelon normal faults die out in the post-rift sedimentary units (de Castro and Bezerra, 2015). In southeastern Australia, a transform fracture zone cuts across preexisting basement structures. Folds and foliations of previous structural stages present unfavourable orientations for reactivation under the present stress field (Lesti et al., 2008).

Preservation of failed rift structures in the geological record depends on the intensity and efficiency of later tectonic and erosion processes. In some cases, the internal architecture and thus the imaging resolution of the basins comprising a failed rift may be lost due to tectonic inversion, severe erosion or even burial into greater depths (Beauchamp et al., 1996; Guiraud and Bosworth, 1997; Beauchamp et al., 1999; Dézes et al., 2004). The reconstruction of the architecture depends on the geophysical imaging resolution (d'Acremont et al., 2005; Enachescu, 2006; de Vicente and Muñoz-Martin, 2013; Melo et al., 2016). The current study focuses on the structural development of a rift front, its failure and later preservation. We concentrate on the Irbid Rift (also referred to as Azraq-Sirhan or Qishon-Sirhan rift) that developed across the Arabian plate and into the Sinai sub-plate during the Oligocene-Miocene (Schattner et al., 2006a; Segev et al., 2014; Fig. 1). Despite tectonic, magmatic and geomorphologic activity post-dating the rifting, the original subsurface structure of the failed rift is preserved at depths of up to 1 km.

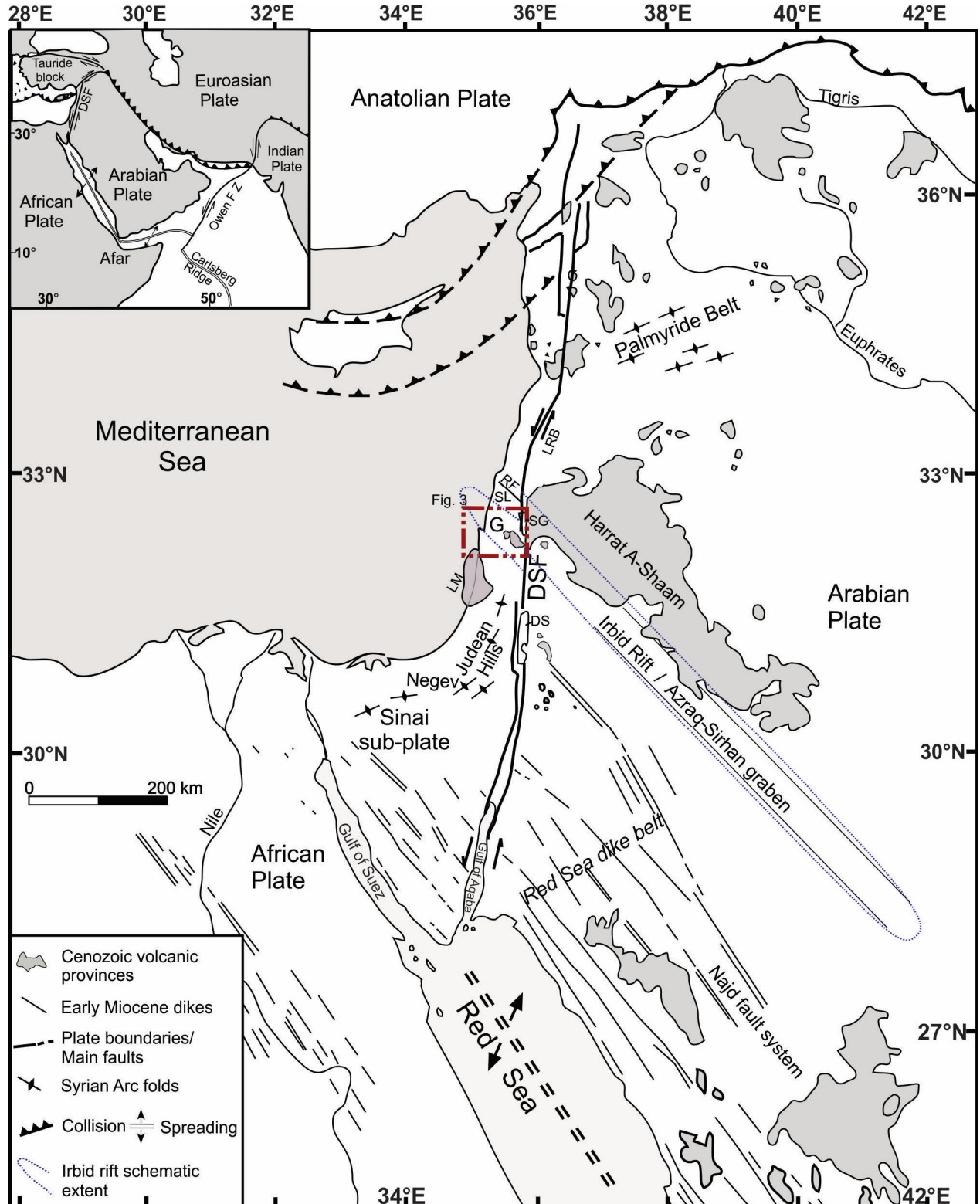

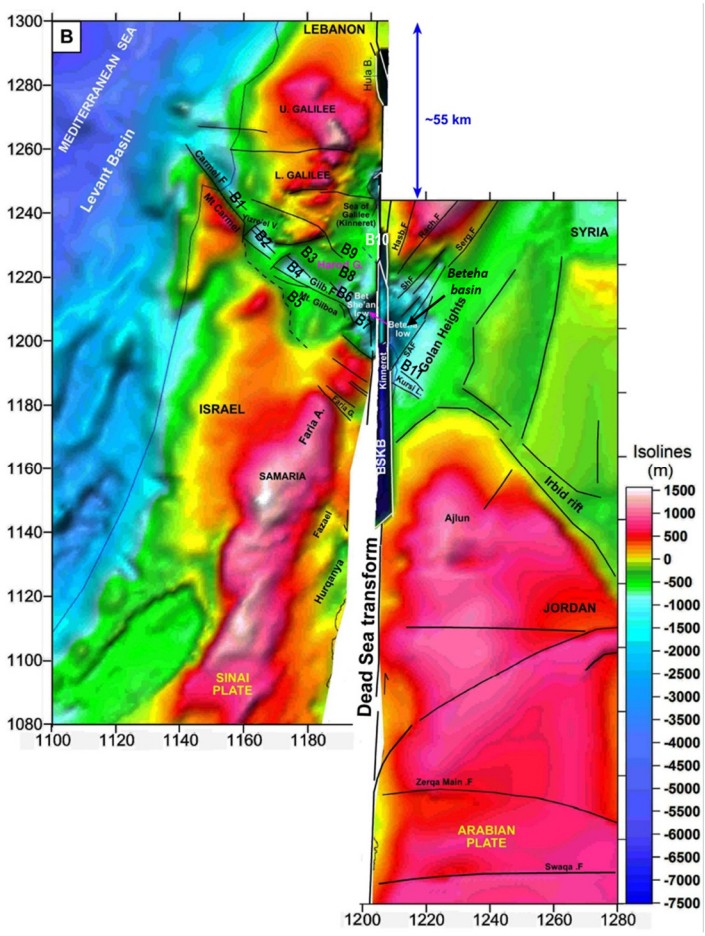

**Figure 1. (A) Major tectonic, magmatic and sedimentary elements along the eastern Mediterranean basin and surrounding plates (after Garfunkel, 1989; Ilani et al., 2001; Schattner et al., 2006a,b; Segev et al., 2014; Segev et al., 2017). Schematic extent of the Irbid rift is outlined in dashed blue lines. Inset- main tectonic elements in the vicinity of the Arabian Plate- the Dead Sea fault (DSF), Afar dome and the Owen fracture zone (FZ). The Cretaceous Syrian Arc fold belt extends from Egypt to Syria across the Galilee. LM- Levant margin; G- Galilee; RF- Roum fault; SL-Southern Lebanon; SG-Sea of Galilee (Fig. 3 for zoom-in); DS- Dead Sea; LRB- Lebanese Restraining Bend. Bordeaux outline- study area, presented in Fig. 3. (B) Reconstructing the pre-DSF plate configuration using the structural map of the top Judea Group interface (modified from Segev et al., 2014). The Beteha basin on the Arabian plate is attached to the Bet She'an basin on the Sinai sub-plate, which showed a ~55 km motion along the DST. Basins referred to herein are marked by the B-series (B1 to B11). Abbreviations: F., fault; V., valley; B., basin; L., low; Mt, Mount; Hasb., Hasbaya; Rach., Rachaya; Serg., Serghaya; SAF, Sheikh Ali fault; BSKB, Bet She'an Kinneret Basin; DST, Dead Sea transform.**

## 2. Regional Geological Setting

The Precambrian basement underlying the Galilee assembled during the Pan-African orogeny until ~620 Ma (Bentor, 1985; Stern, 1994; Stein and Goldstein, 1996; Stern and Johnson, 2010). Subsequent truncation eroded a 6-10 km thick section from the Galilee area (Garfunkel, 2002). The Paleozoic opening of the Palmyride rift (overlapping the current location of the

Palmyride Belt, Fig. 1) crossed the Galilee in an NNE orientation (Walley, 1998; Segev and Eshet, 2003). Opening of the Levant Basin (Garfunkel and Derin, 1984; Robertson, 1998; Garfunkel, 1998, 2004; Gardosh et al., 2008) during the early Cretaceous (Segev et al., 2018) re-defined the formerly inland Galilee region as a new continental margin. The passive margin accumulated marine sediments until the Late Cretaceous.

Progressive closure of the Neotethys Ocean at the northern Arabian plate (Stampfli and Hochard, 2009; Frizon de Lamotte et al., 2011) induced compressional stresses across the Levant margin. The stresses inverted the extensional grabens formed 100-200 m.y. earlier and folded the Levant margin (Sagy et al., 2017). A ~50 km wide S-shape fold belt developed from northern Sinai, through Israel, and along the Palmyride region (the 'Syrian Arc'; Krenkel, 1924; Hensen, 1951; Guiraud and Bosworth 1997; Walley, 1998; Hardy et al., 2010). Compressional stresses kept the margin at shallow depths, while the syn-tectonic

chalks of the Santonian-Paleocene Mount Scopus Gr. covered the late Cretaceous relief. During the Paleogene-Eocene tectonic and thermal quiescence led to gravitational vertical subsidence of NW Arabia. The resulting transgression submerged the entire Galilee under more than 1000 m of ocean water. Lower-middle Eocene sediments comprise mainly chalks and limy chalks with sporadic chert nodules and chert layers (Sneh et al., 2000a; Segev et al., 2011).

Mantle upwelling of the Afar plume began at the late Eocene, uplifting the overlying crust (Hofmann and Curtillot, 1997; Pik

et al., 2003; Avni et al., 2012 and references therein). Part of the mantle plume volume propagated away from the plume head northwards during the Oligocene-early Miocene (~25-17 Ma). Its imprint on surface topography was recorded as a gradual and continuous uplift migration across northeastern Africa, gradually exposing the region above sea level. The exposure led to a regional truncation that levelled the area into a low-relief peneplain over merely 7 Ma (e.g. Egypt, Jordan, southern and central Israel; Picard, 1943; Picard, 1951, Quennell, 1958; Garfunkel and Horowitz, 1966; Garfunkel 1970; Horowitz 1979,

1992, 2001; Ben David and Mazor, 1988; Zilberman 1989, 1992; Avni 1991, 1993, 1998; Ben David, 1993; Bar et al., 2013, 2016 and Avni et al., 2012). In the Galilee, the Regional Truncation Surface (RTS) serves as a marker, dividing between marine carbonates below and lacustrine, fluvial and volcanic rocks above (Picard, 1943; Wald et al., under review; Wald et al., 2014; Wald, 2016). Meanwhile, Eocene chalks and Paleocene-early Miocene greenish-gray shales and marls accumulated on the Levant margin (Fig. 2; Gvirtzman et al., 2011; Steinberg et al., 2011).

The Afar uplift was accompanied by a regional crustal extension and formation of two NE-SW trending coeval rifts (Schattner et al., 2006a). The NW trending Red Sea-Suez rift divided Arabia from Africa (Steckler and ten Brink, 1986; Bosworth et al., 2005), while the NW trending Irbid Rift developed across the Arabian plate. The northwestern front of the Irbid rift crosses the southern Galilee (Fig. 1; Shaliv, 1991; Schattner et al., 2006a). The Irbid rift divides two crustal terranes that differ in thickness and seismicity (Ginzburg et al., 1994; Hofstetter et al., 1996; Ben-Avraham et al., 2002; Segev et al., 2006), and

possibly form two different sub-plates (Palano et al., 2013; Schattner and Lazar, 2014). A series of basins subsided along the NW propagating Irbid rift. They developed across the present-day Galilee, up to the Levant continental margin (Lyakhovsky et al., 2012; Segev et al., 2014). However, unlike the Red Sea, spreading across Irbid rift failed to mature into a young ocean (Shaliv, 1991; Schattner et al., 2006a). The Galilee basins subsided during the late Oligocene-Miocene terminal stages of Irbid

rift. They maintained their low topographic relief despite intense tectonic activity along the nearby Dead Sea Fault plate boundary (Shaliv, 1991; Matmon et al., 2003).

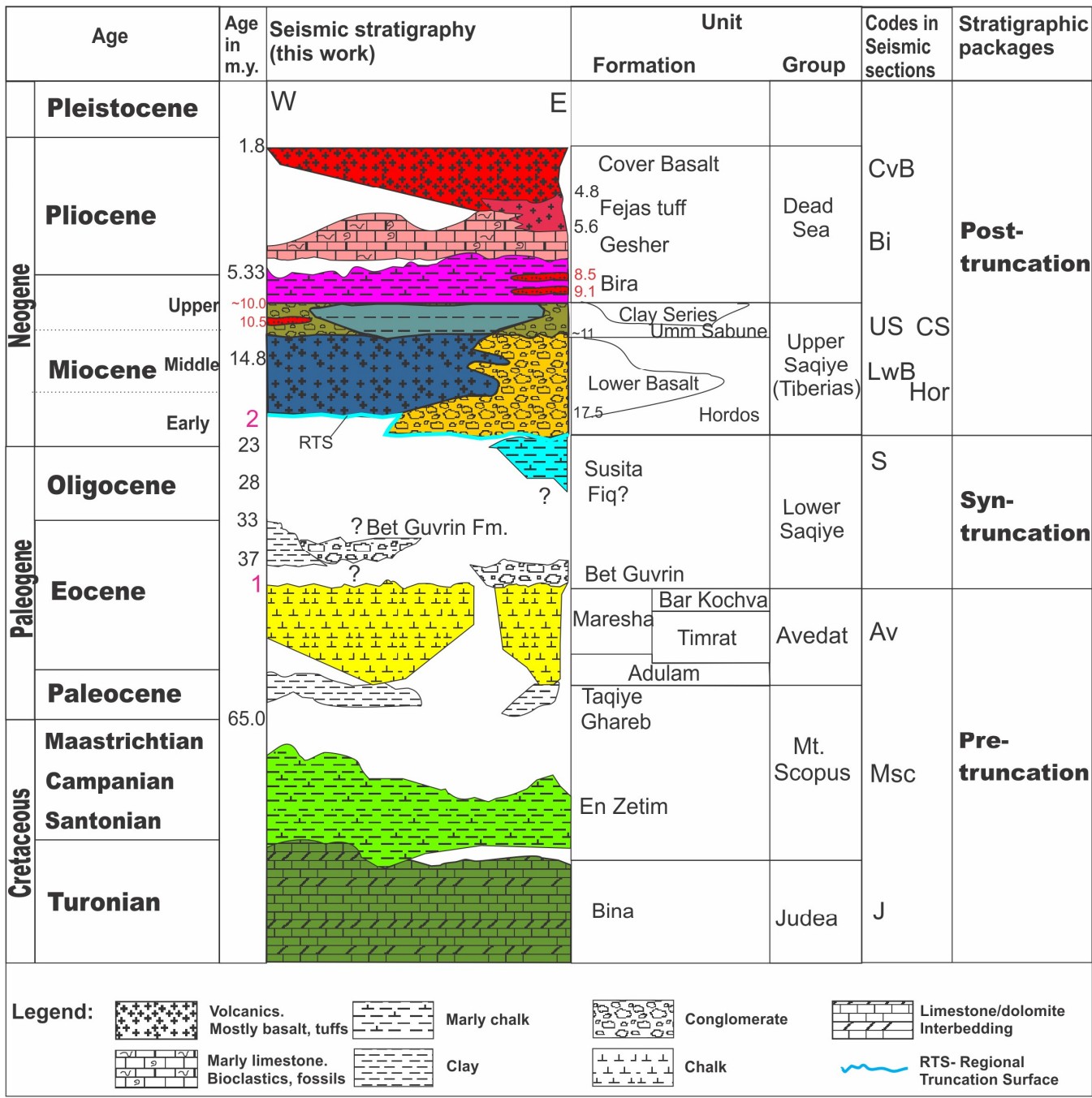

Figure 2. Stratigraphic correlation across the Galilee. Pink numbers (1, 2) represent unconformity surfaces. Radiometric ages (in million years, i.e. m.y.) from Shaliv, 1991; Heimann, 1996; Segev, 2000. Dating of Tuff Fejas, Gesher and Cover Basalt formations

**(shown in red) from Rozenbaum et al. (2016). Base of Cover Basalt age is from site 3 of Dembo et al. (2015). CvB-Cover Basalt Formation.; Bi-Bira Formation; US-Umm Sabune Formation; CS-Clay Series; S-Susita Formation; Av-Avedat Group; MSc-Mt. Scopus Group; J-Judea Group.**

Lateral motion along the N-S trending Dead Sea Fault (DSF) plate boundary initiated between 18 Ma (Freund, 1970; Garfunkel, 1998, 1981; Joffe and Garfunkel, 1987) and 14 Ma (Bayer et al., 1988; Bosworth et al., 2005). In a recent study, Nuriel et al. (2017) dated the onset of motion along the DSF. Their calcite age-strain analyses yielded ages of 20.8-18.5 Ma for the southern DSF, and 17.1–12.7 Ma for the DSF in northern Israel (next to our study area). The motion decapitated Irbid rift and isolated the Galilee Basins on the newly formed Sinai sub-plate (Schattner et al., 2006a). Transtenssion along the DSF

resulted in further subsidence of basins along it during late-Miocene-early Pliocene (Garfunkel, 1981; Joffe and Garfunkel, 1987; Smit et al., 2010). Around 5 Ma the lateral displacement along the DSF reached ~40 km, while extension across the valley was ~4 km (Joffe and Garfunkel, 1987). However, since 5 Ma subsidence of basins along the DSF accentuated (e.g. Gulf of Aqaba, Dead Sea, and Hula basins; Figs. 1,3; Garfunkel and Ben-Avraham, 2001). This trend was also recorded in the basins situated at the junction between the DSF and Irbid rift trends (Figs. 1, 3): Bet Shean (B7), Kinarot (B10), and Sea of

Galilee basins (Hurwitz et al., 1999; Segev et al., 2014; B7, B10 in Fig. 3). Further north, increased transpressional motion along the DSF (Freund, 1970; Schattner and Weinberger, 2008; Weinberger et al., 2010) uplifted the Lebanese restraining bend (Fig. 1, e.g., Walley, 1998; Gomez et al., 2006; Gomez et al., 2007). Contraction of the bend induced an N-S extension of the Galilee basins. As a result, the formerly Irbid rift basins remained low in both structure and topography (Schattner et al., 2006a, b).

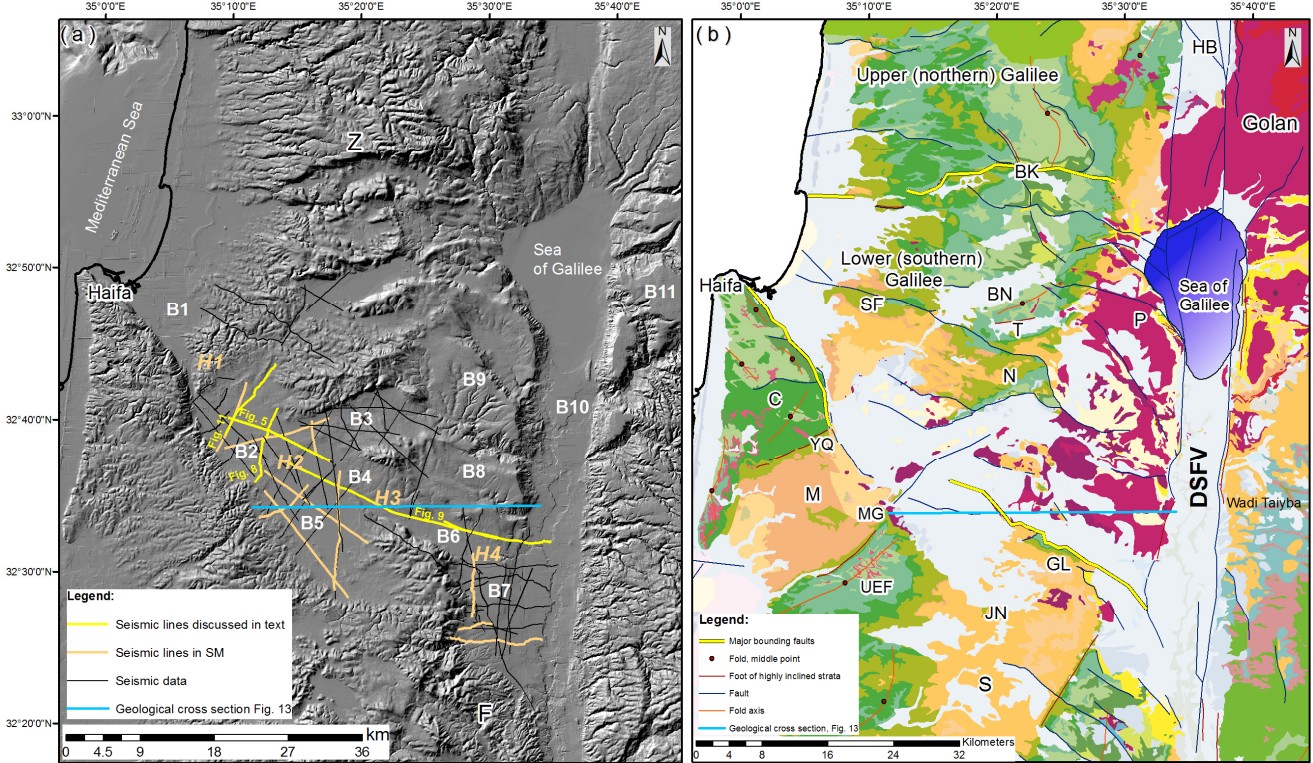

**Figure 3. Location map of the study area, Sinai sub-plate. (a) Location of the multi-channel seismic reflection profiles used in this study on a shaded relief digital elevation model (DEM- Sneh et al., 2000b). Local names of the basins and the structural highs are abbreviated to simplify the description, as follows: B1- Zevulun basin, B2- Yizre'el basin, B3- Kesulot basin, B4- Afula basin, B5- Taanach basin, B6- Harod basin, B7- Bet Shean basin, B8- Moledet basin, B9- Sirin basin, B10- Kinarot basin, B11- Southern Golan basin. H1-Tivon hills, H2- Hayogev-Mizra horst, H3- Navot high, H4- Sede Nahum high. (b) A 1:200,000 geologic map (Sneh et al., 1998). Note major faulted boundaries: the Carmel (C)-Gilboa (GL) southern fault boundary and the Zurim escarpment northern fault boundary. For color code see Fig. 2, with two exceptions - Lower Basalt Fm. (purple in map) and the Neogene formations: Hordos, Clay Series, Bira, Gesher- all of which appear in crème. Abbreviations: F-Fari'a anticline, Z-Zurim escarpment, dividing between upper and lower Galilee, BN- Bet Netofa, BK- Bet Hakerem, P- Poriyya, T- Tur'an, SF- Shefar'am, TVN- Tivon, N-Nazareth, C- Mt. Carmel, M- Menashe syncline, S- Shekhem syncline, YQ- Yoqneam, MG-Megiddo, UEF- Umm El Fahm anticline, JN-Jenin, GL- Mt. Gilboa, DSFV- Dead Sea Fault Valley, HB- Hula Basin.**

## 3. Morpho-tectonics of the southern Galilee basins

The southern Galilee Neogene basins extend across ~50 km, between the DSF and the Levant continental margin (Fig. 3). The Carmel-Gilboa (C, GL in Fig. 3b) and Zurim fault (Z in Fig. 3b) systems in the south and north (respectively) bound the Southern Galilee basins (Schattner et al., 2006a, b). Their N-S extent narrows westwards from ~35 to ~10 km in a low relief that exhibits sporadic highs dividing local valleys. The surface of westernmost basin, Yizre'el (B2) lays at 30-70 m above sea level. To the east Kesulot (B3) and Taanach (B5) basins are at 60-100 m, Harod basin (B6) is between 30 and -210 m, and Bet

Shean (B7) is at -250 m. The low relief of the southern Galilee basins (i.e. valleys and intervening small hills), divides between two segments of the Mesozoic Syrian arc fold belt (Krenkel, 1924; Fig. 3b). The remnant Mesozoic Syrian arc fold belt, that currently builds the Israeli hilly backbone, raised by ~500 m since the Pliocene (Fig. 3b- GL, C, UEF, N, SF). Lower Cretaceous (Kurnub Group) and Jurassic (Arad Group) exposures appear in limited areas. The upper Cretaceous Judea and

Mount Scopus Groups are exposed mainly along the fold belt truncated crests, for example along the Gilboa, Carmel, and Nazareth ridges). The fold belt synclines are also uplifted, to ~250 m, exposing the Eocene Avedat Group (across Tivon and Menashe hills; Fig. 3).

Sedimentary infill of the southern Galilee basins comprises intercalations of siliciclastic, volcanic and carbonate lithologies of the Dead Sea and the Upper Saqiye (previously Tiberias) Groups (Fig. 2). They accumulated mainly under continental

(lacustrine-fluvial) conditions with phases of shallow marine intercalations. Since the early Miocene and until the present, the relatively high rims of the basins have contributed clastics that accumulated in the basins (Shaliv, 1991; Sandler et al., 2004; Rozenbaum et al., 2016). This mixture resulted in a discontinuous and irregular distribution of sedimentary units and facieses across the southern Galilee basins. Some of the units wedge laterally (e.g., Um Sabune Conglomerate Fm., Bira Fm. in Fig. 2) while others appear only locally.

A series of studies conducted over the last half century provide invaluable insights into the stratigraphy, hydrology, geophysics and outcrop mapping of the study area and its surroundings. They include masters and PhD theses as well as reports and peer-reviewed papers (e.g., Schulman, 1962; Sass, 1966; Yair, 1968; Weiler, 1968; Dicker, 1969; Klang and Sherman, 1972; Dekel, 1988; Shaliv, 1991; Hatzor, 1988; Gev, 1989; Sneh et al., 1998; Gardosh and Bruner, 1998; Bartov et al., 2002; Rotstein et al, 2004; Sagy and Gvirtzman, 2009; Segev et al., 2006; Abelson et al., 2009; Zilberman et al., 2009). Some of the studies focused

on volcanism, paleo-drainage, and paleohydrology of the Yizre'el basin (Yair, 1968; Schulman, 1962; Wishkin, 1973; Shaliv, 1991; Gev, 1989; Baer et al., 2006). Geophysical studies showed the architecture of basins along the southern Galilee: Bet Shean basin (Meiler et al., 2008; Gardosh and Bruner, 1998; B7 in Fig. 3); Zevulun basin (Sagy and Gvirtzman, 2009; B1 in Fig. 3); Taanach and Yizre'el basins (Politi, 1983; Rotstein et al., 2004; B5 and B2 in Fig. 3). These studies, focused on localized structures across the southern Galilee basins, left the larger, regional, context unresolved. The current study integrates

all the previous results with unpublished data to address fundamental questions regarding the origin and development of the lower Galilee. It surveys the geometry of the basins to clarify regional structural relationships: is it a single continental basin that accumulated sediments from its surrounding rims (Picard, 1943; Schulman, 1962; Shaliv, 1991)? Alternatively, maybe a full graben bounded by longitudinal faults, Zurim and Carmel-Gilboa from the north and south respectively (as suggested by Kafri and Ecker, 1964; Mero, 1983) or possibly a couple of half grabens bounded by these faults (as proposed by May 1987;

Matmon et al., 2003)? What is the structural and tectonic association between the southern Galilee basins development and the nearby DSF and Levant continental margin? More specifically, what is the relationship between the southern Galilee basins and the Irbid rift? In what manner does the structural development of the southern Galilee basins relate to the regional volcanic events?

## 4. Dataset and Methodology

Geological reconstruction of the structure and development of the southern Galilee basins relies on an integrated interpretation of all available geophysical and geological datasets from the study area. The new database was constructed on the Kingdom Suite (IHS) platform. It includes 70 multi-channel seismic reflection profiles, 506 boreholes, outcrop data, and previous seismic
interpretations. The seismic reflection data were acquired between the 1970's through 2000's. The profiles cover a total length of 800 km. The average depth imaging is 500-1000m below the seismic datum (sea level). The boreholes depth ranges between 35-2390 m below surface. Seismic resolution enables the interpretation of geological units starting from the upper Cretaceous (Fig. 2).

Stratigraphic, hydrological, geophysical and outcrop datasets collected in the past across the study area are integrated here into
a single database in a WGS 1984-UTM 36N datum-projection, bridging over gaps in vertical and horizontal resolution, reflector amplitudes, processing methods and datum. These sources include Schulman (1962), Sass (1966), Aizenberg (1967), Yair (1968), Weiler (1968), Dicker (1964), Dicker (1969), Klang and Sherman (1972), Dekel (1988), Shaliv (1991), Hatzor (1988), Gev (1989), Sneh et al. (1998), Gardosh and Bruner (1998), Bartov et al. (2002), Rotstein et al (2004), Sagy and Gvirtzman (2009), Segev et al. (2006), Abelson et al. (2009), Zilberman et al. (2009). Data were further used for constructing
structural maps of key surfaces. The surfaces and faults were exported from the Kingdom Suite to Petrel (Schlumberger) to build a structural model. Results of previous geological mapping were used to extend the structural model from sea level datum (elevation of 0m) up to the present-day topography (30-550m asl). Two velocity surveys were done in the area (Sarid 1 and Revaya 7 wells; SM3, 7, 9). The synthetic seismogram of Revaya 7 well (Frieslander, 1997; Meiler et al., 2008) enabled a reliable stratigraphic correlation with the seismic data. In addition, using a 2000 m/sec velocity for the shallow, near-surface
beds (weathered beds), enabled correlation between depth and time domains. Completion of the structural model relied upon digitization of truncation surfaces from previous studies in ArcMap (ESRI) (Weiler, 1968; Dicker, 1969; Dekel, 1988; Shaliv, 1991; Shaliv, 2003; Sneh, 2008). Outcropping truncated surfaces are considered as layers within a specific unit rather than its top (due to erosion). Control points were added from boreholes. Integration of all datasets yielded a coherent database and a three-dimensional geological grid model of the Galilee subsurface, extending from a depth of 2500m to the present-day surface
topography.

## 5. Results

The results section describes the sedimentary fill of the basins in chronological order. It is followed by a description of the structural elements. Local names of the basins and the structural highs (i.e. uplifted blocks) are abbreviated to simplify the
description (Fig. 3). All geographical location mentioned in text appear in the Google Earth™ supplementary material, herein referred to as GE. The sedimentary fill is bounded between two temporal and structural markers. The basin floor is marked by the Oligo-Miocene Regional Truncation Surface (RTS; ~23-17 Ma; Figs. 2, 4), a peneplain predating the subsidence of the

basins. The RTS truncates the folded and displaced structures of the Judea, Mt. Scopus and Avedat Groups (Fig. 2). The latter is thinning towards H2 and pinches out approximately 400 meters west of it (Fig. 5). An important surface culminating the Neogene sedimentary fill is the top of Bira Fm., depicting a very mild relief. The Cover Basalt Fm. locally covers it and provides a temporal marker. Analysis of the entire database indicates that the type section is located along the axis of the Southern Galilee basins (B2, B4, B6, and B7). Basin depocenters align along a northwest axis (Figs. 1B, 7). Further details from B3, B5, B8-B9 basins complete the section. Additional information from B1, B10, and B11 is provided in the discussion (location: Fig. 3).

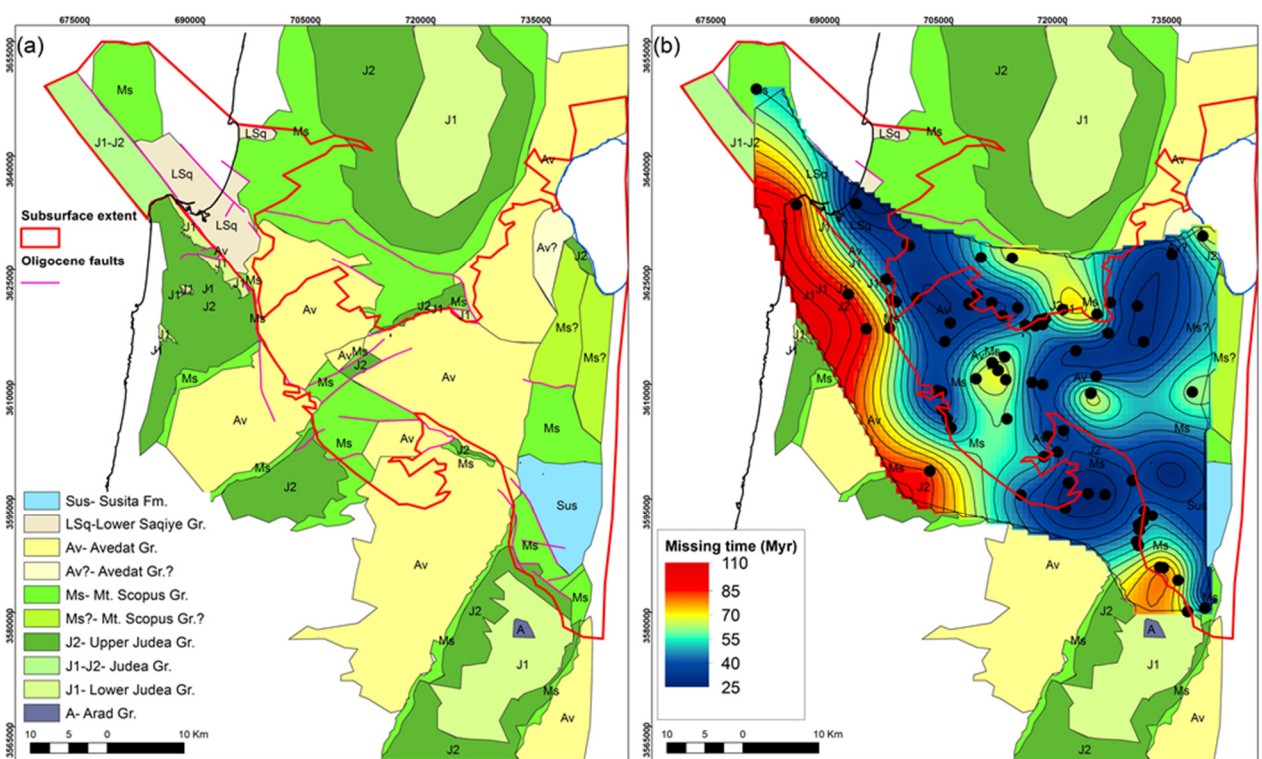

**Figure 4. (a) Subcrop map of the Oligocene regional truncation surface (RTS) in UTM GWS1984 Zone 36N projection. The map shows the youngest units truncated by the RTS, based on the integrated interpretation of geological and geophysical data, from the surface and subsurface. Colors correspond to the seismic profiles. Red polygon marks the extent of data gathered in the current study. (b) Spatial variation in truncation across the Galilee is a product of kriging interpolation, further represented by contours of equal time gap in million years. Black dots mark locations where the youngest unit below RTS and older unit from above are available for quantifying the time gap (the time gap is discussed in Wald et al., under review). Some of the data points are today exposed above the datum of the map. Note that some of the points may include pre-Oligocene truncations.**

## 5.1 Basin fill

The oldest formations deposited above the RTS are the contemporaneous Lower Basalt and Hordos Fms. (Fig. 2). Today, these formations appear in the subsurface, and also outcrop across marginal areas and local highs (Fig. 6). The Hordos Fm. predates

the Lower Basalt Fm., yet their seismic appearance is similar. They resemble in reflection frequency, amplitude, and continuity. Some differences between these formations appear in parts of B7. Seismic and borehole data (Fig. 9, Figs. S5, S8, S14) show that the Hordos Fm. covers the floor of the B6-7, 10 basins, and thickens southwards along the DSF. Further up the section, it interfingers with the Lower Basalt Fm. that thins southwards along the DSF. The Lower Basalt Fm. directly overlies the basin
floor in B2-5, excluding local highs (Figs. 5, 7). The concordant seismic appearance of the formation hints to the consecutive succession of basalt flows, and the hiatuses between them (Figs. 5, 7, 8). The lateral continuation of reflectors degrades towards fault and fold zones, representing displacement events postdating the accumulation of the Lower Basalt Fm. (Figs. 9, 10, 11; Figs. S1-4, S8, S10). The Lower Basalt Fm. is missing from B1, where the oldest basin fill unit comprises marls associated with Bet-Guvrin Fm. (Lower Saqiye Group; Figs. 3, 4).

Numerous seismic, borehole and outcrop datasets indicate that the Lower Basalt Fm. generally thickens towards the center of each of the basins (Fig. S6). The thickening is also indicated by the arrangement of main faults, dikes, and volcanic feeders (Figs. 5, 10, 11, Figs. S1-5). In B3-7 and H2 the thickness exceeds 100 m. In B2, the Lower Basalt Fm. fills a Cretaceous syncline while onlapping its flanks. It thickens from a few meters over H1 to a constant ~125 m at the center of B2. The thickness of the Lower Basalt Fm. reaches 400-600 m adjacent to H2 (Fig. 10). At the western part of B4, a borehole crossed
630 m of the Lower Basalt Fm. (Table 1). However, this is a minimal value since the base of the formation has not been reached. B3 is divided into two sub-basins by H2. The eastern part of B3 accumulated 50-100 m of Lowe Basalt Fm., while the western part accumulated at least 350 m (base of the formation was not reached). In the eastern border of B4 and B5, the Lower Basalt Fm. reflectors onlap an elevated Eocene block (H3) at ~10°. The Lower Basalt Fm. thickness does not exceed 200 meters in B5. Its reflectors appear parallel/subparallel to the basin floor (RTS, Figs. S3, S16). Further east, near B6-7
Lower Basalt Fm. thickness varies considerably between 395 to 750 m (Table 1). In B10 the Lower Basalt Fm. reaches 3500 m (Table 1). The southern subsurface limit of the Lower Basalt Fm. is Nahal Bezek fault, whereas a localized several tens of m thick outcrop appears further south in Marma Fayad (location: Fig. 6, Google Earth Archive- GE; Figs. 6, 7, 10a).

Top of the Lower Basalt Fm. is an erosional unconformity that accentuates eastwards, according to the age of the units overlying it (Figs. 2, 3, 6, Fig. S5). In the west, Um Sabune conglomerate and the Clay Series Fms. overlay the Lower basalt
in B2-5 basins (Figs. 5, 8, 11). Bira Fm. covers this unconformity over the H2, H3 structural highs and across B6 (Fig. 9). In the eastern Galilee (B8-9) and B7, the top Lower Basalt unconformity is either directly overlain by the Cover Basalt Fm. at elevated terrains (e.g. Yisachar-Gazit and Hashita-Geva blocks of B8, location: Fig. 10a) or covered by the Bira Fm. (Figs. 3, 10, Fig. S8).

| Data source (Well name, Seismic data, reference) | Associated basin | Thickness (m) | Base reached? | Basin floor | Figure |
|---|---|---|---|---|---|
| Poriyya type section; Shaliv (1991); Schulman (1962) | B10 | 750 | No | Senonian | Figs. 8, 9 in Shaliv (1991); |

| | | | | | location: Fig. 3, 10b, GE |
|---|---|---|---|---|---|
| **Gideon 5** | B4 | 630 | No | Senonian | Fig. 5, GE |
| **Bira 3** | B8 | 450 | No | Eocene | GE |
| **Shadmot Devora** | B9 | 385 | Yes | Eocene | GE, Fig 6 |
| **Belvoir 1** | B6-7 | 660 | Yes | Senonian | GE, Fig 6 |
| **Seismic data** | B7 | 1000 (interfingers with Hordos Formation) | Yes | Senonian | Figs. 9,11 |
| **Inbar, 2012** | B8 | 2000-3500 | No | Senonian | |

**Table 1 - marked thicknesses of Lower Basalt Fm.**

The clastic formations of the Dead Sea Gr. overlie the truncated top of Lower Basalt Fm. (Figs. 8, 9, 11, Figs. S1-6, S8, S10,

S14). Data indicate that the group accumulated during the upper Miocene-Pliocene in a lacustrine/fluvial environment. Appearances of lumachelle ostracods at the Bira Fm. indicate an episodic connection to the marine environment. Interchanging paleosol horizons and volcanic remains crossed in boreholes point to exposed continental environments. Um Sabune Conglomerate Fm. overlies Lower Basalt Fm. at H1, the margin of B2 (Kishon 1 borehole, Fig. 8, GE), and in the eastern Galilee. The conglomerates appear near the margins of the basins and volcanic centers. They are bounded by the intersection

between Gevat and Nazareth faults (Fig. S7). Um Sabune Conglomerate Fm. contains basaltic pebbles derived from the Lower Basalt Fm., as well as alluvial carbonate and basaltic pebbles that experienced extensive mechanical reworking.

The Clay Series Fm. is contemporaneous to Um Sabune Conglomerate Fm. (Figs. 2, Fig. S7). The grain size of both formations decreases upwards as well as towards the depocenters of each basin. The geographic coverage of these formations defines the present spatial extent of basins B2-6 (Fig. 3). The Clay Series Fm. appears at the center of B2-B7. In places, it directly overlays

the Lower Basalt Fm. (e.g., Taanach 4 borehole, Fig. S3, GE). Its thickness is relatively constant along the axis of the central basins B2 (400 m), B4 (200 m) and it reduces towards B6. In more peripheral areas it ranges around tens of meters (Figs. 8, 9, 11, S1-6, Table 1). The thickness differences may point to differential subsidence while deposition.

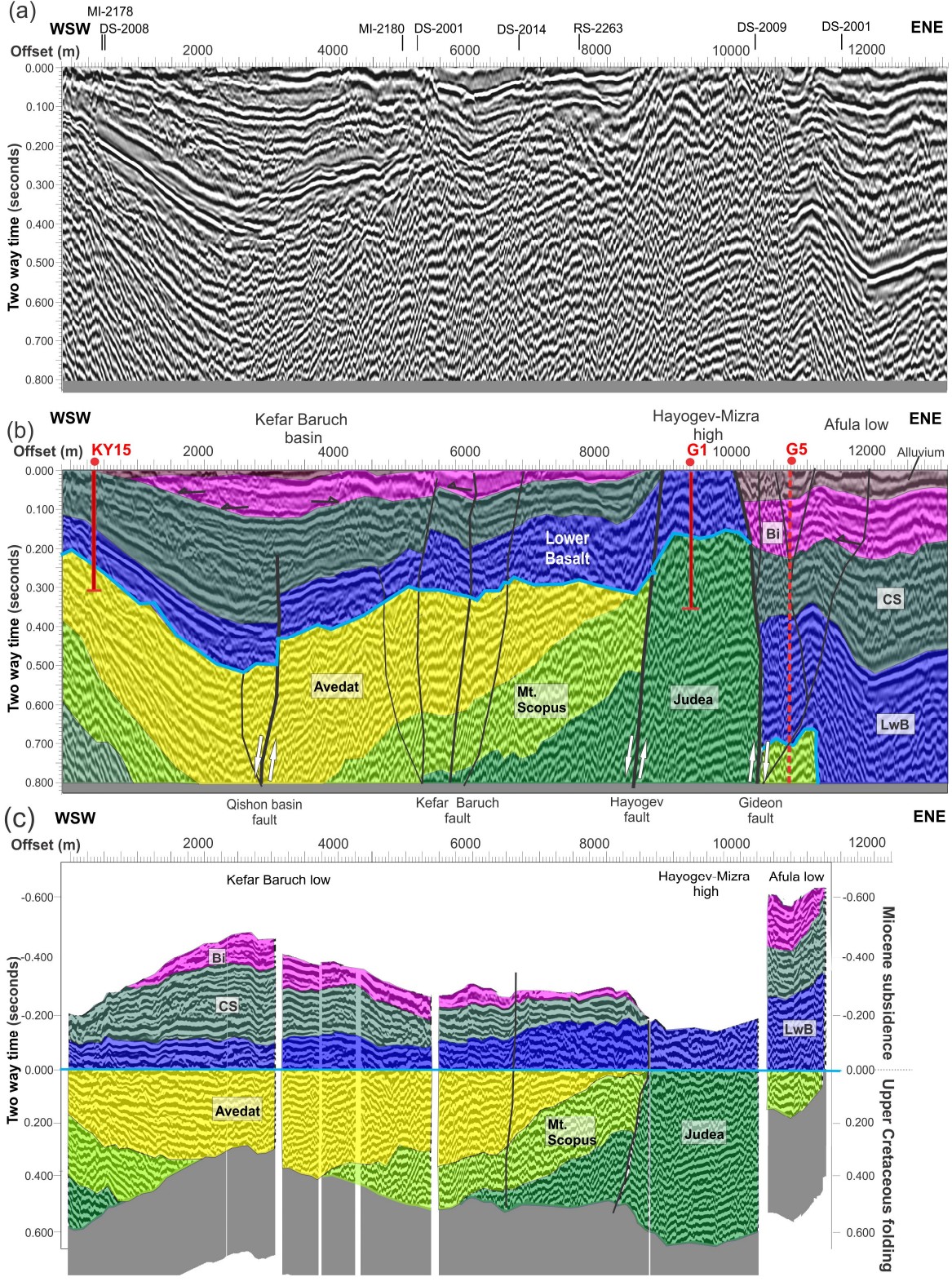

**Figure 5. (a) Multi-channel seismic reflection profile line MI-2187 crossing the basin axis (location: Figure 3). (b) The RTS horizon (celeste) divides pre-truncation from post-truncation sediments. Hayogev-Mizra Horst (HMH) intervenes between Kefar Baruch and Afula Neogene basins. Cretaceous units at the syncline were interpreted using intersecting and overlapping deeper seismic profiles from the DS series (see Fig. S3). Boreholes KY15- Kefar Yehoshua 15, G1-Gideon 1 and G5- Gideon 5, projected by 1 km from south (location: Fig. 3b, GE). Uppermost unit (gray): alluvium. (c) Same profile, flattening of the celeste horizon (RTS) to image the truncation. The flattening tool enables a comparison between predating and postdating sedimentary stacks. Flattening the RTS in the seismic software hints at RTS predating and postdating main processes. For example, in Kefar Baruch basin, Cretaceous folding shown by a syncline- predates the RTS, while Neogene subsidence, shown by an accumulation of Neogene sediments- postdates the RTS. Bi-Bira Formation; CS-Clay Series; LwB-Lower Basalt Formation; Groups: Avedat Group; Mt. Scopus Group; Judea Group. Vertical exaggeration: x5.**

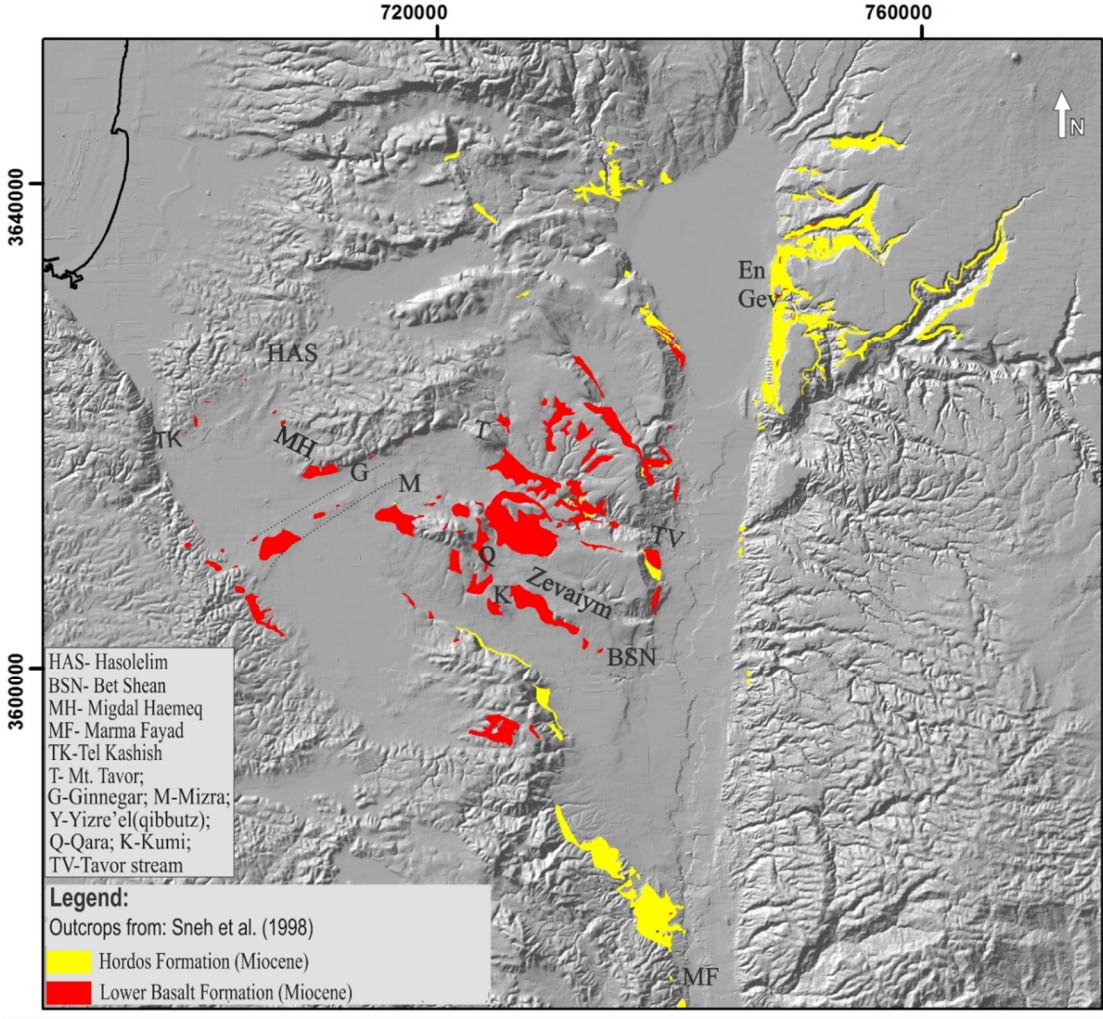

**Figure 6. Location of Lower basalt Formation and Hordos Formation outcrops. The westernmost outcrop is along the eastern margins of H1 (location: Fig. 3; DEM- Sneh et al., 2000b).**

Lower Basalt Fm. is covered by three younger formations: Bira Fm., Gesher Fm. and locally by the Cover Basalt Fm. (Fig. 2). Seismic resolution does not allow to differ between the Bira Fm. and the Gesher Fm. so these two units are generally termed Bira Fm. in seismic profiles shown here. The Bira Fm. consists mostly of marls, but also of marine and lacustrine limestones, gypsum and salt. Its thickness ranges between 0-200 m (Fig. 9, Fig. S5). Bira Fm. also overlies Um Sabune Conglomerate and

5   Clay Series Fms. in places (Figs. 2, 5, 8-9, 11, Figs. S1, S4-6). In seismic data Bira Fm. appears as a continuous set of reflectors, detectable across the basins (Figs. 5, Fig. S4) even in folded and faulted regions (Figs. 8, 11). Reflectors at the base of the formation onlap an unconformity (Figs. 5, 8, Figs. S2, S4). The top of Bira Fm. is an unconformity surface (Fig. S4). In places, it is overlain with paraconformity by the Cover Basalt Fm. (Fig. 9). Bira Fm. is missing over topographic and structural highs (Figs. 5, 9, Fig. S5).

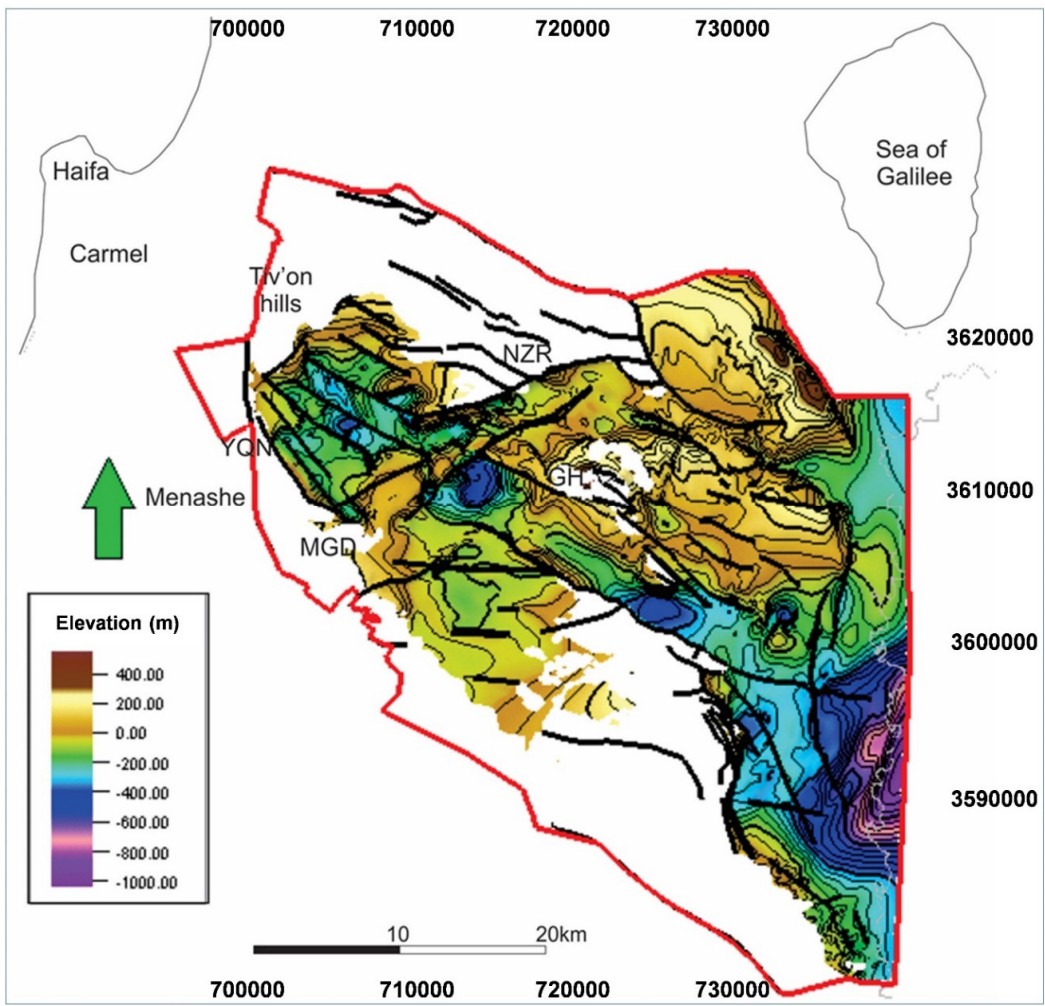

**Figure 7. Structural map of the top of the Lower Basalt Formation surface. Note that the lowest areas strike NW. GH- Givat Hamore, MGD- Megiddo, YQN- Yoqneam, NZR- Nazareth.**

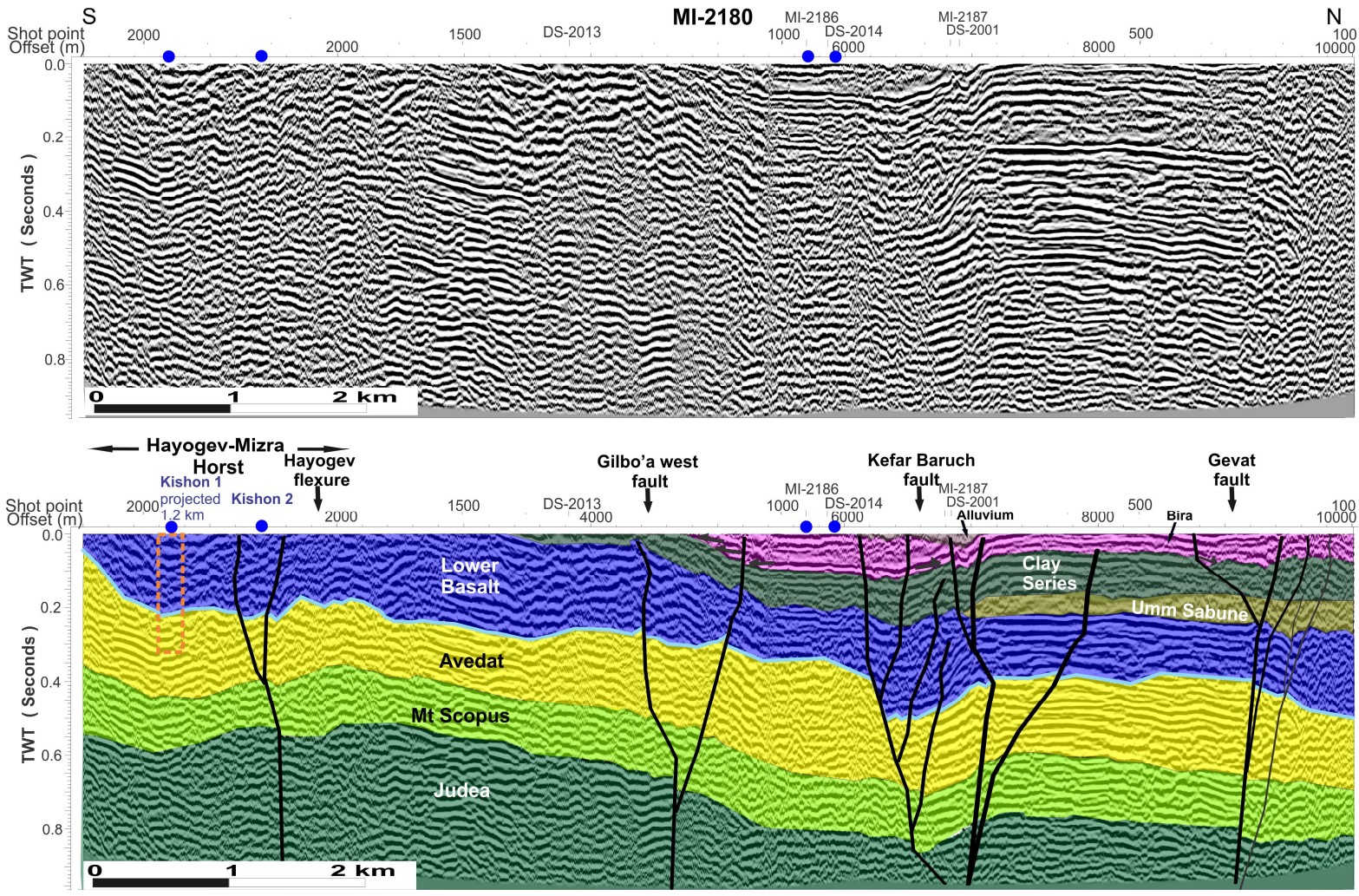

**Figure 8. Multi-channel seismic reflection profile line MI-2180. Kefar Baruch basin (B2) and Hayogev-Mizra horst (HMH) are sheared by faulting and folding. Vertical offset alongside folding on fault branches deform the Clay Series, Um Sabune and Bira formations. Gevat fault suggests a horizontal offset due to its near vertical fault plane and 1 km wide flexures (see also Fig. 11). Thick Lower Basalt formation on the south suggests a volcanic source in HMH area (see also Fig. 10). This profile cuts the primary deformation zone and its uplifted southern shoulders- HMH. Celeste horizon- RTS. Orange dashed rectangle- projected location of the Kishon 1 well. Arrows depict onlap of the Bira Fm on the Clay Series. Vertical exaggeration: x2.5.**

**5.2 Faults**

Three types of faults appear in the database: (1) major marginal faults that bound the southern Galilee basins from north and south; (2) faults dividing between basins, sub-vertical to the basin axis. Their orientation ranges between NE to NNE; and (3) Through-going faults that cross the basins. The current study focuses on the first two types, while the third is at the center of Wald et al. (under review).

**5.2.1 Major faults**

Three major marginal faults define the southern rim of the Southern Galilee Basins. In the NW, the Carmel fault down-throws B1 by ~1500 m. Further ESE, a series of normal faults, includes the Yoqneam fault, whose downthrown side is B2. The throw decreases southeastwards from ~200 m to ~50 m (Figs. 7, 10C, GE). The trace of Yoqneam fault diminishes to the SE until it intersects with Gideon and Hayogev faults in Megiddo region (western margin of B4-5; Fig. 8). The Umm El-Fahm fold plunges NE towards B5, where it appears at a depth of 150-200 m below surface (Fig. S9). Given the poor seismic imaging, a southern bounding fault is marked as suspected (Figs. 10b-c, Figs. S10, S11). However, this discontinuity of reflectors may be ascribed to an apparent structural throw, termed Dotan flexure herein (Figs. 10c, 13, Fig. S10) between Umm El-Fahm anticline (Fig. 3, Figs. S10, S11) and Shekhem syncline (Fig. 3, Fig. S9), of the upper Cretaceous Syrian Arc fold belt (Fig. 3).

The amount of displacement increases again along the Gilboa fault in the southeast. The Gilboa fault extends from the middle of H2 southeastwards (Fig. 10, Figs. S9, S11). In the NW the Gilboa fault appears in the subsurface of northern B5, where the entire package of reflectors of the basin fill is dipping northwards, towards B4. It downthrows B4 by 400 m relative to B5. The fault downthrows B6 about the Gilboa block footwall (Figs. 7, Fig. S11, GE). The fault is detectable across the shallow subsurface, up to the seismic datum (mean sea level), and exposed in places. This suggests it was active at least through the Plio-Pleistocene. In the east, the Gilboa fault also appears in the subsurface of B7, where it forms a flower structure, attesting to a lateral component of displacement. Vertical displacement along the fault is in the range of 100 m (Fig. 10, Fig. S11). In the southeast, Tayassir, Bardala, and Bezeq faults bound B7 from the south (Figs. 10, Figs. S9, S11, S14). These faults divide between the basin and the NNE trending Faria anticline that plunges from the south. At the eastern boundary of the study area, DSF truncates the eastern part of B7 (Fig. S14).

The northern border of the Galilee basins is the E-W trending Bet Hakerem fault system (including Zurim escarpment) and Ahihud fault (e.g., Matmon et al., 2003; Schattner et al., 2006b; Figs. 3, 10c). The Neogene basins mapped here pinch out northwards and do not reach these faults. Therefore, the E-W trending Tur'an, Bet Netofa and Bet Hakerem valleys are excluded from the current analysis (locations: Fig. 3). A series of NW to W trending faults divides between the latter E-W valleys and the Neogene basins. The western segment of Bet Qeshet fault borders H1 from the north. Further east, three step faults downthrow B2 (Zarzir, Timrat, Nahalal faults; Figs. 10c, 11, Fig. S2). The NE-trending Nazareth fault downthrows B3

southwards, while B3 fill is dipping to the north (Fig. S4). East of B3, the Tavor horst (T in Fig. 6) is uplifted along the eastern segment of Bet Qeshet fault (Figs. 3, 10c, 12, GE). The fault divides the horst from the Sirin-Qama block (B9- Fig. 3, location of fault: Fig. 10c, GE). Neogene exposures extend up to the northeastern corner of the southern Galilee basins (Fig. 6, 10). However, in this area, the delimitation of southern Galilee basins is less clear, due to later displacements.

### 5.2.2. Secondary faults

A series of NNE to NE-trending normal faults divide between the basins and structural highs of the southern Galilee. The faults are nearly perpendicular to the axis of the basins complex. Seismic data show that displacements across these faults are mainly vertical with a horizontal component. Regional numerical modelling of Lyakhovsky et al (2012) followed by a review of rift-transform interaction adjacent to continental margins (Segev et al., 2014), has predicted rift-perpendicular features. Locally, these faults, structural highs, and basins between them are evident from the structural map of top Avedat Gr. that consist the floor of most of the basins (Fig. 12). The following paragraphs describe the division along the major axis, from NW to SE.

The structural and topographic transition between H1 and B2 occurs along a lineament associated with Sede Yaakov and Aloney Abba faults. These faults are derived from the geological map (Sneh et al., 1998; Segev et al., 2006) since a seismic profile does not intersect them. These faults expose fragmented outcrops of the Lower Basalt Fm., as well as a chain of localized springs (Figs. 3, 10c, GE). The intersection between Sede Yaakov and Gilboa West faults in the WSW of B2 is a fracture zone (Tel Kashish; location: GE; Figs. 7, 10, 11, 12, Fig. S12). Hayogev fault bounds B2 in the east, defining the transition to the NE-trending H2. The Lower Basalt Fm. forms a westward dipping monocline above the fault (Fig. 5).

The H2 horst is topographically elevated by several tens of meters above B2 and B4. H2 plunges to the NE into the subsurface of B3, partially dividing B3 into two sub-basins (Figs. 5, 6, Fig. S1, GE). Plio-Pleistocene sediments are absent from the top of H2. The Lower Basalt Fm. overlies an erosional unconformity of the top Judea Fm. (Gideon 1 and 4 wells; location: GE; Figs. 5, 9) and Mt. Scopus Gr. (Gideon 3 well; Fig. 5; Figs. S3, S4, S10; location: GE). Gideon fault bounds H2 from the east, down-throwing B4. Normal displacement along this fault is ~100 m in its northern and southern margins. It reaches ~500 m in the middle (main axis of the basins). Correlation between seismic data and Gideon 1, 2, and 5 wells (Figs. 5, 9, GE, Fig. S5) show uneven thickness between the fault flanks, suggesting that it was active several times during the mid and late Miocene, at least until the end of deposition of Bira Fm. (Figs. 5, 7, 9, Fig. S5).

Three structural elements separate B3 from B4. Afula fault vertically throws Lower Basalt Fm. reflectors northward by app. 200 m (Figs. 7, 10). East of the fault the volcanic Givat Hamore and Ein Dor blocks separate B3 from B4. (Fig. 10, location: Fig. 7, GE). Gideon 5 well located along the margin of B4 crossed 980 m of Neogene basin fill and did not encounter the base of Lower Basalt Fm.. This suggests that vertical displacement across Gideon fault occurred during the mid-Miocene. The displacement took place concurrent with dike intrusions and uplift of Givat Hamore and Ein Dor blocks (Figs. 7, 10).

Gilboa fault defines the boundary between B4 and B5 to the south, off the axis of the southern Galilee basins. Data indicate that the B5 fill thickens northwards towards Gilboa fault (Fig. 12, Fig. S16). B5 is bounded by H2 in the west and H3 in the

east. Avital fault crosses the NW corner of B5 (Fig. 10c, Fig. S10). Displacements along this sub-vertical fault are mainly horizontal. They are associated with branching into secondary faults and local folding (Figs. S10, S16).

The elongated B6 basin extends along the main axis of the southern Galilee basins, north of the Gilboa fault. The Lower Basalt Fm. covers the WNW margin of H3. An intermediate graben hangs as a step between B6 and H3, faulted along Gilboa fault (Shaliv 2003, 2005). Seismic data show that the northern limit of B6 is downthrown along Hashita and En Harod faults relative to Hashita-Geva/Zevayim block (Figs. 7, 10, 12, 13). Sub-vertical normal faults downthrown B7 relative to the eastern flank of H4 and Hashita-Geva block (Figs. 9, 13; Figs. S11, S13). Bet Shean fault is the easternmost of this series. It downthrows the Lower Basalt Fm. 200 m on its eastern side (Fig. S14). However, the basin fill thickens and tilts to the east, where its original structural boundary is unclear. Similarly, the structural transition from B7 northwards into B8 is vague.

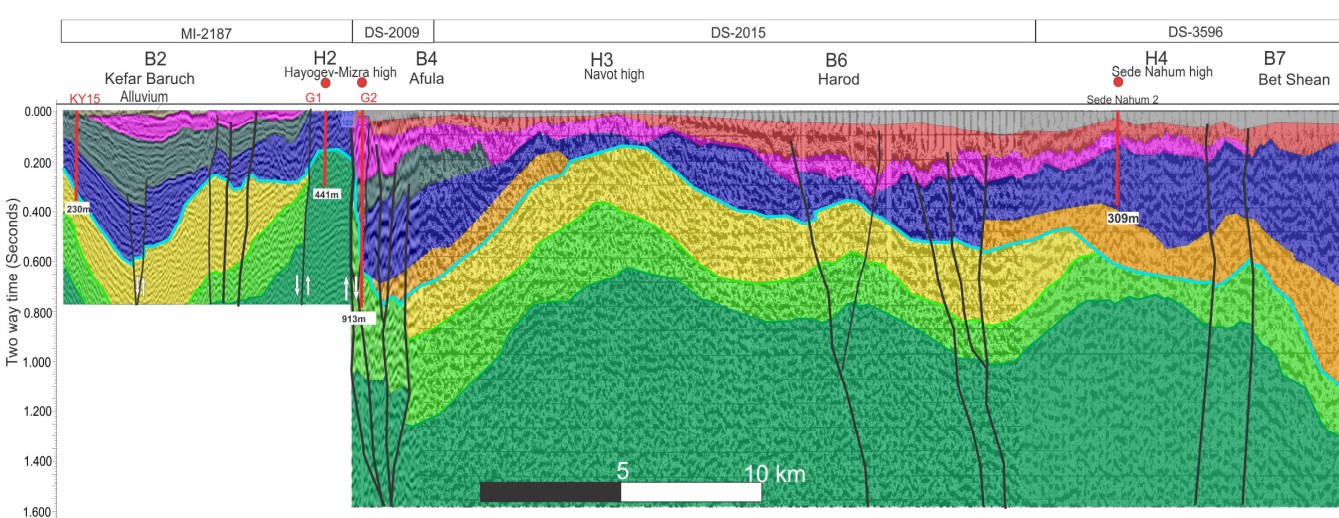

**Figure 9. Multi-channel seismic reflection profile across basins B2, B4, B6, B7 (lines MI-2187, DS-2009, DS-2015, DS-3596). The RTS horizon (celeste) divides pre-truncation from post-truncation sediments. Hayogev-Mizra High (HMH) intervenes between Kefar Baruch and Afula Neogene basins. Cretaceous units at the syncline (Kefar Baruch, B2) and in the eastern B7 area were interpreted using intersecting and overlapping deeper seismic profiles from the DS series. Location: Figure 3; Unit color code- Figure 2. No vertical exaggeration.**

## 6. Discussion

Integrated analysis of the geological-geophysical dataset shows the structural development of the original flat Oligocene to early Miocene RTS (Fig. 2; Avni et al., 2012) into a series of extensional basins (grabens and half grabens). The discussion addresses the development of the basins based on their structure and stratigraphy. It then suggests a classification of the

southern Galilee basins, at each stage, given the regional tectono-stratigraphic events and comparison to similar structures worldwide. These insights are used for understanding the structural development of a failing rift during its final stages.

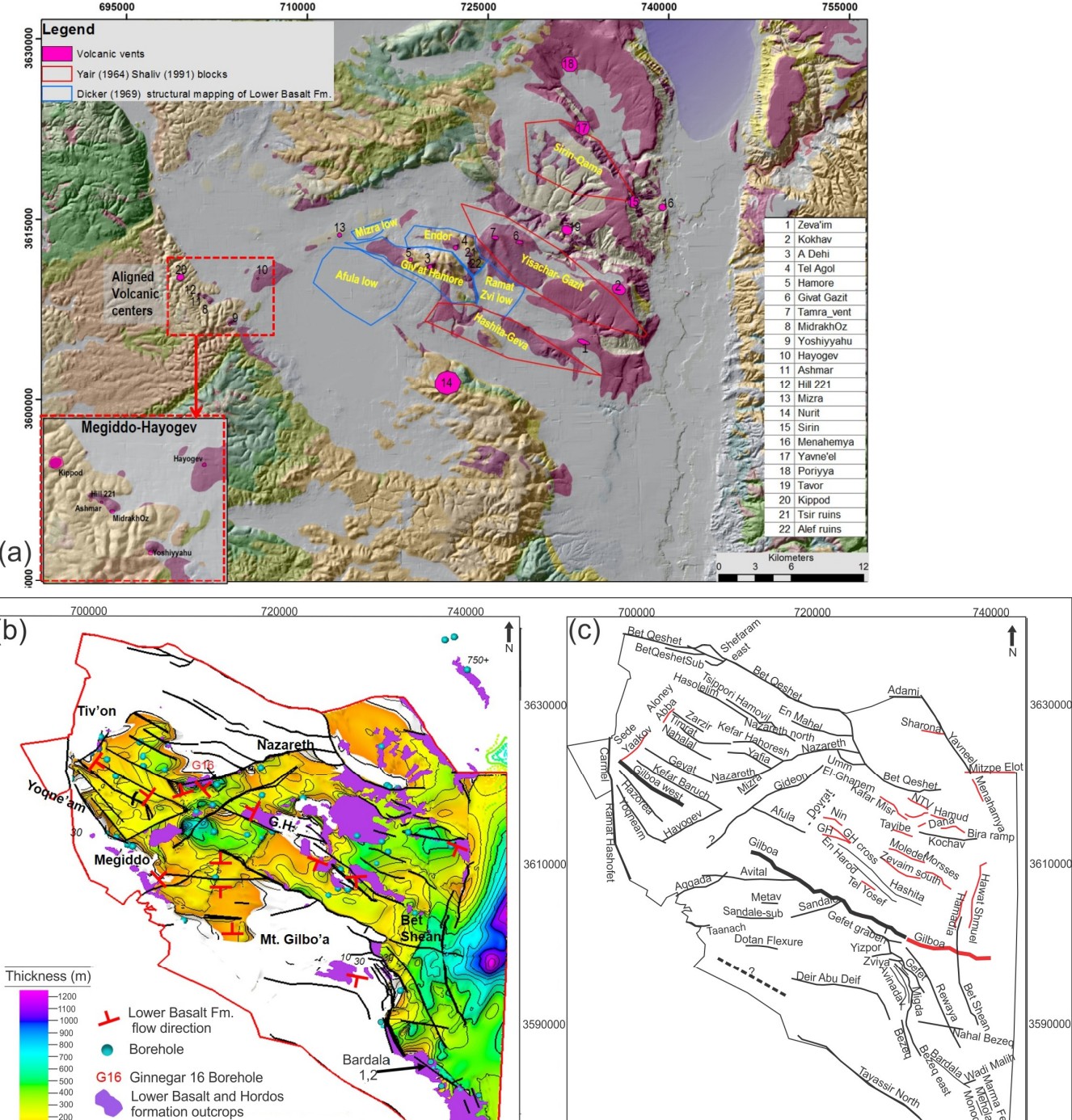

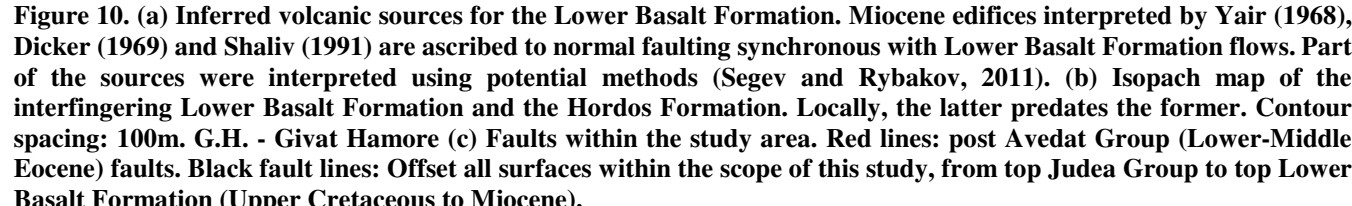

**Figure 10. (a) Inferred volcanic sources for the Lower Basalt Formation. Miocene edifices interpreted by Yair (1968), Dicker (1969) and Shaliv (1991) are ascribed to normal faulting synchronous with Lower Basalt Formation flows. Part of the sources were interpreted using potential methods (Segev and Rybakov, 2011). (b) Isopach map of the interfingering Lower Basalt Formation and the Hordos Formation. Locally, the latter predates the former. Contour spacing: 100m. G.H. - Givat Hamore (c) Faults within the study area. Red lines: post Avedat Group (Lower-Middle Eocene) faults. Black fault lines: Offset all surfaces within the scope of this study, from top Judea Group to top Lower Basalt Formation (Upper Cretaceous to Miocene).**

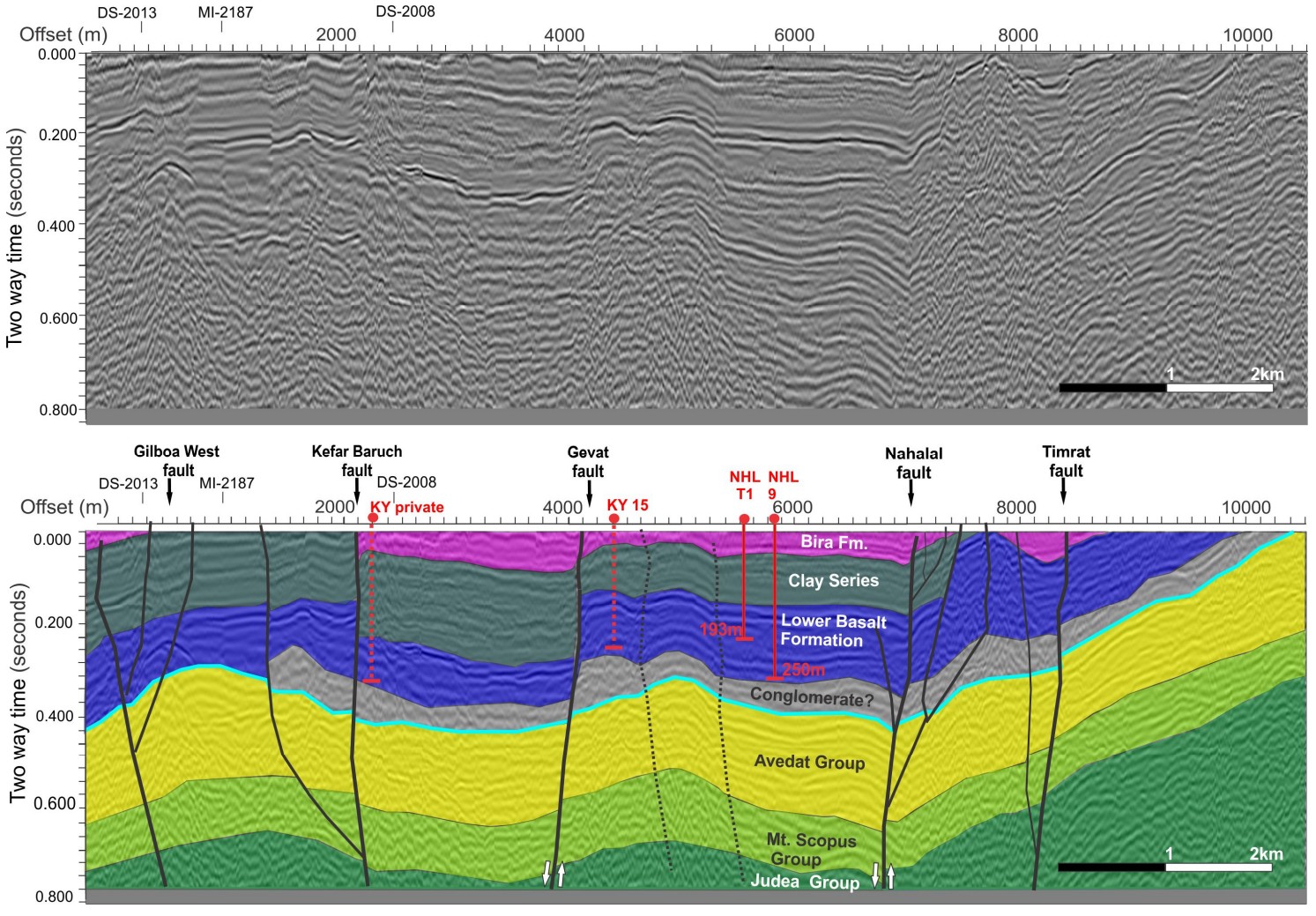

**Figure 11. Multi-channel seismic reflection profile line MI-2178. Strike-slip faults with a normal component in the frame of Plio-Pleistocene lateral shear adjacent to Mt. Carmel and Tivon blocks. Normal faulting vertically offsets basin fill units. Post-Bira Formation folding (postdating uppermost Miocene-Pliocene) is assigned to the strike-slip shear on the originally normal faults. A paleo alluvial fan, predating the vertical offset, is depicted by the Clay Series. Celeste horizon- RTS; dashed lines- projected wells. NHL- Nahalal; KY- Kefar Yehoshua. Location: Figure 3; Unit color code- Figure 2. Vertical exaggeration: x3.**

**6.1 Subsidence of basins**

**6.1.1. First stage (20-9 Ma)**

The first stage of subsidence initiated during the early-mid Miocene. The subsidence occurred mainly near the eastern part of
the southern Galilee basins, across B6-11 (Fig. 3). Subsidence and faulting developed while the conglomerate member of
Hordos Fm. accumulated in topographic lows (Schulman and Rosenthal, 1968; Garfunkel, 1989). A composite section crossing
the basins along a WNW trajectory shows that Hordos Fm. accumulation in B6-8 was accompanied with normal faulting and
folding (Fig. 9). However, remains of Hordos Fm. are not restricted the subsiding basins. Their extent is larger than the current
northwest array of basins. They appear in sporadic outcrops, such as Marma Fayad and Ein Gev (thickness exceeds 200 m;
location: Fig. 6, GE); in various elevations on the northern flank of Faria anticline; across the tilted blocks of the eastern
Galilee; and across southern B9. Above mentioned evidence suggest that the current shape of the southern Galilee basins was
formed by younger deformations, while preceding Miocene basins extended further south of their present-day structure. It also
indicates that these remains were displaced by younger faults (Figs. 10, 12; Shaliv et al., 1991) that were active during the
initiation of motion along the DSF (Freund, 1978; Garfunkel, 1981; Garfunkel, 1989).

The Spatial and temporal provenance of the lower to mid-Miocene conglomerate of Hordos Fm. are still debated. Conglomerate
accumulation of the Hordos Fm. suggests that basin subsidence predates the Lower Basalt Fm., although in several localities
it inter-fingers with it (Fig. 9, Figs. S8, S14). Temporal emplacement therefore is tricky. Outcrop and seismic data from B2
show that normal faults displace a conglomerate unit, before the Lower Basalt Fm. accumulated (Fig. 11). Sandler et al. (2004)
associates the conglomerate unit to Bet Nir Fm., suggesting it is concurrent with the Lower Basalt Fm. (17-9 Ma). Our
integrative morpho-structural analysis bridges over the spatial gap between the isolated patches of the conglomerates (e.g.
Kafri (2002) provenance study), suggesting that Bet Nir and Hordos Fms. accumulated at the same time frame. Together they
are products of the same paleo-drainage system that extended from the east to the west across the low relief of the Galilee,
immediately before the subsidence of the basins.

The southern Galilee basins accumulated an up to 650 m thick section of volcano-clasts and flows of Lower Basalt Fm. during
their subsidence (Fig. 10). In general, the thickness of a basaltic unit is expected to increase close to its source. This assumption
guided the identification of volcanic sources across the study area. The seismic and borehole database provided evidence for
thickness variations and information about the lithology. Previous studies provided basalt dating from outcrops and wells,
along with mapping of tilted blocks and faults (Fig. 10; GE; Segev et al., 2006; Dicker, 1964; Schulman, 1962; Shaliv, 1991).
Integration of the data sources indicate that the basalts arrived through dikes (e.g. Gilboa, Mishmar Haemek), stocks (Givat
Hamore), volcanic eruption centers (Kippod, Kochav Hayarden, Tel Agol), and fault planes (Sede Yaakov, Moledet, Yoqneam,
Sandale, Aloney Abba; Figs. 7, 10, 11, 12). Baer et al. (2006) dated the eruption at Givat Hamore to 13.5 Ma. Geochemical
analysis of volcanic products suggests that the lithosphere of the Galilee has been rich with veins that fed the Miocene

magmatism (Weinstein, 2000). Some of the volcanic sources (e.g., dikes, Hazor, 1988; Shaliv, 1991) follow the southern boundary faults of the basins, suggesting a possible connection (Figs. 7, 10).

During the mid-Miocene, normal displacements along faults facilitated deepening of the basins (Figs. 7, 10c, 12). Structural signature of the left-lateral displacement along the DSF enhanced between 12-14 Ma. Bosworth et al. (2005) suggest that the movement started at ~14 Ma in association with the transition of Red Sea opening. In response, the slip along DSF shifted from a N60°E opening motion, perpendicular to the Red Sea axis, to a N15°E motion, diagonal to that axis but parallel to the axis of the DSF. Others estimate the initiation of DSF displacement in the study area to 13 Ma (Shaliv, 1991). Northward channelling of the Afar plume (Ritesma et al., 1999; Chang et al., 2011; Hansen and Nyblade, 2013) along with geodetic and structural research (Bellahsen et al., 2003; Bosworth et al., 2005; ArRajehi et al., 2010) suggest a transition in stress regime. Three-dimensional analogue models of the Red Sea-Gulf of Aden rift system point at an increase of 70% in the rotational relative motion between Africa and Arabia since 13 Ma (Molnar et al., 2017). This pronounced shift at 13 Ma has left footprints in the Galilee branch.

The association between volcanism and tectonics specifically around 13 Ma appears in several studies across the Arabian plate (e.g., Bayer et al., 1989; Camp and Roobol, 1992; Ebinger and Casey, 2001). Until 13 Ma volcanic activity closely follows the faulting event. A marked shift in volcanism is noted at ~13 Ma. In the western Arabian plate, volcanic fields renewed their activity after a cessation of 9 Ma (Bohannon et al. 1989; Camp and Robool 1992; Ilani et al. 2001; Krienitz et al. 2009). In contrast, magmatic activity in the Galilee was relatively continuous. K-Ar dating bound the volcanic activity across B2 between 16-9 Ma (i.e., the Lower Basalt Fm.; Shaliv, 1991). Further to the east across B6-B11, H3 and Mt. Gilboa, older K-Ar ages of 17-15 Ma were retrieved (Shaliv, 1991; 3,5,14, 19 in Fig. 10a). Updated 40Ar/39Ar dates yield a lower limit of 17 Ma for the Lower Basalt Fm. (Rozenbaum et al., 2016; Sandler et al., 2015). Since 13 Ma, volcanism was active across Harrat-A-Sham-western Arabia and the Galilee. It was active during the subsidence of the southern Galilee basins and accumulation of conglomerates.

Integration of all above evidence indicates that during the first stage an E-W trending paleo-drainage system developed across the southern Galilee, accumulating conglomerates. Shortly after, this drainage pattern ceased during the relief accentuation due to subsidence of a series of <10 km wide grabens and half-grabens. The basins collected conglomerates, separately, along with the Lower Basalt Fm.. The basins subsided along an NW-trending axis (Fig. 12). Within this general trend, some individual basins trend to the WNW and W. These basins continued to sink, extend and even merge during the transition to the second stage of subsidence.

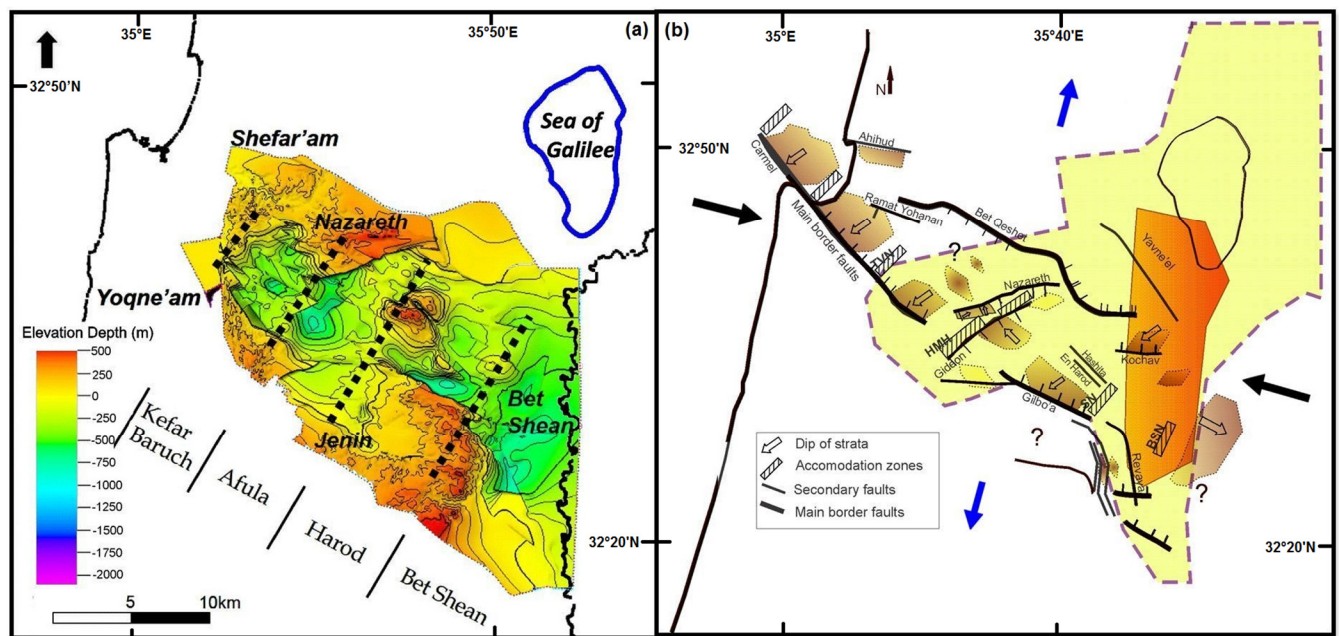

**Figure 12. Structure of the study area during the Cenozoic. (a) Structural map of the top Avedat Group. Dashed lines**
**portray the structural highs (H1-4 in Fig. 3a). The current structure of the Galilee is a product of two main subsidence**
**phases shown in b. (b) Aerial extent of the first subsidence stage (20-9 Ma) is outlined in Light yellow. Dark orange-**
**pronounced subsidence during the first stage, overlapping the current Dead Sea fault valley.  The second subsidence**
**stage (9-5 Ma) is outlined by a series of NNE trending basins, perpendicular to the major axis of basins from the first**
**stage. Darker brown- pronounced subsidence. The second stage stress field is depicted by black arrows (compression)**
**and blue arrows (extension). Bounding normal faults also exhibit a lateral component. HMH- Hayogev-Mizra high,**
**TVN- Tivon, SN- Sede Nahum, BSN- Bet Shean.**

### 6.1.2 Second stage (9-5 Ma)

Tectonic displacements that acted during the first stage of subsidence continued during the second, along with erosion. A series

of blocks and depressions depicted from the structural map of the Lower Basalt Fm. points at the continuance of vertical

motions. Basins continued to subside, forming local topographic lows that accumulated the erosion products. Conglomerates

of the Um Sabune Fm. settled close to the edges of the basins (Fig. 8, Figs. S2, S7). Their composition includes pebbles of

Lower Basalt Fm. as well as older carbonates (Sandler et al., 2004). Grain size of the conglomerates decreases upwards

(Schulman, 1962), indicating a moderation of tectonic activity along the rims of the basins with time. Um Sabune Fm. outcrops

tilt southwards along the northern rim of B2 (Kafri, 2002); consist 200 m of the Shokek 1 well, drilled in a western marginal

graben of B7, unconformably covering Avedat Fm. (location- GE); occur at the northern plunge of the Faria fold, southern B7 border (Shaliv et al., 1991); compose the upper part of B8-9 inter fingering with the Bira Fm. (see below). The Um Sabune Fm. appears to thicken within the incised channels that drain B6. The thickening could result from two factors. Syn-tectonic magmatism allowed the Lower Basalt Fm. to accumulate within subsiding basins on the one hand whereas other parts of the formation were uplifted across their rims. The basins deepened while their margins were gradually elevated (Dicker, 1964). Therefore, elevated terranes and basinal margins were the provenance of the Um Sabune Fm.. Ongoing subsidence of B7-8 during the mid-late Miocene facilitated the accumulation of thick section of Um Sabune Fm. near the margins of the basins (e.g., Bet-Yosef, Neve Ur and Zemach wells, Fig. 8; Fig. S7; locations: GE), the Clay Series Fm. was deposited within their depocenters (B2-6; Figs. 8, 9, 11; Figs. S1, S2, S4-6). The Clay Series Fm. has been preserved since most of the tectonic activity focused on the edges of the basins. According to tens of water wells this formation is verified as a local aquiclude (Wishkin, 1973).

Deposition of the Bira Fm. occurred during the volcanism that produced the Intermediate Basalt Fm.. This volcanic formation mainly follows faults (Shaliv, 1991) and due to its minor occurrences (thin sections of few to tens of meters) seismic resolution does not permit its interpretation. It occurs cross H3 (Shaliv et al., 1991), along Rewaya and Gefet faults (Fig. 9, Figs. S5, S8, S14) in B7-10 and the central Jordan Valley (Schulman, 1962; Rozenbaum et al., 2016). With time, accumulation of Bira Fm. moderated the rugged relief of the Galilee until it became almost flat at the end of the Miocene (Fig. S5). The outcrops of Bira Fm. appear today close to faults that were active during the second stage of subsidence, and in places cover these faults. This evidence suggests that Bira Fm. recorded the cessation of subsidence of the southern Galilee basins. The cessation might be associated with a short-term tectonic quiescence across Sinai sub-plate and its nearby Levant margin, allowing marine transgressions to cover the low relief of the southern Galilee.

Previous studies suggest that part of Bira Fm. accumulated across the southern Galilee basins during one or more marine transgressions during the upper Miocene (Blake, 1935; Schulman, 1962). Shaliv (1991) suggests the transgression occurred between 7-6 Ma (Tortonian), whereas the global eustatic record does not contradict additional marine intercalations between 5.4-5.25 Ma (e.g., Haq et al., 1987; Müller and Hsu, 1987). This deduction is also supported by marine mega-fauna (Shaliv, 1991), ostrea lumashell unusual facies in outcrops of southern B9 (Schulman, 1962) and lithological resemblance of the latter evidence and those of the southern Galilee basins marine succession (Michelson and Lipson-Benitah, 1986). The transgressions probably arrived from the west (Mediterranean) since at that time the topographic valley along the N-S trending DSF already existed (Fig. S14; Segev et al., 2017). In addition, lithology of Bira Fm. shows a distinct marine to estuarine (saline to brackish) facies shift from west to east (Dicker, 1964). The change occurs north of B6 (B7-8, along Moledet-Bira 2-Bira 4-Shadmot Devora wells, see GE). Gvirtzman et al. (2011) describe a lateral facial shift during the late Miocene (Fig. 13 in Gvirtzman et al., 2011): Pattish Fm. represents the first facies of a continental shelf (i.e., marine) environment. The transition to the second lacustrine floodplain facies of Bira Fm. is located on the eastern flank of H1, next to the intersection between Sede Yaakov and western Gilboa faults in Tel Kashish (Figs. 3, 10c, Fig. S12; Zilberman and Sandler, 2013). Further east, the third facies of the Bira Fm. is represented by the land locked lake environments of the Sedom Fm (i.e. along the Dead Sea fault).

During the late Miocene, Tortonian episodic marine transgressions filled the southern Galilee basins. Saline conditions developed in separated water bodies as evident from the accumulation of laminar marls and evaporates (Bira and Sedom Fms; Shaliv, 1991; Fig. 17d in Segev et al., 2017). Clean gypsum crystals found at the outlet of Tavor stream are associated with proximal lagoon depositional environment (location: Fig. 6; GE). Rozenbaum et al. (2016) suggest that the gypsum crystals formed before the onset of the Messinian salinity crisis, i.e. prior to 5.96 ± 2 Ma (Krijgsman et al., 1999; Manzi et al., 2013). Our data show that chalks and limestones were deposited in shallow basins at the Yisachar and Poriyya area, while conglomerates accumulated along the rims of the basins (Fig. S15). This flooding is contemporaneous with the onlap of the Pattish Fm. reefal limestones along the Israeli coastal plain.

At the end of the second stage, shallow brackish water lakes occupied the topographic lows above the basins. Limestones and chalks of the Gesher Fm. accumulated in lakes (Shaliv et al., 1991; Rozenbaum et al., 2016). The thickness of Gesher Fm., merely reach tens of meters, slightly above the seismic resolution limit. Bira and Gesher Fms. sealed the southern Galilee basins and formed a relatively flat relief. Similar to the RTS at the base of the basins, the relatively flat top of the Bira and Gesher Fm. serve as a marker for tectonic activity that deformed the study area during the Plio-Pleistocene.

Data presented in this study suggest that the uplift of Mount Carmel, Tivon and Shefar'am occurred close to the end of the second stage, between 5-6 Ma (Figs. 3, 12). The uplifts placed topographic barriers between the Mediterranean Sea and the inland lakes, diverting possible marine transgressions to regions south of the Galilee. These observations stand in line with Shaliv (1991). Gvirtzman et al. (2011) suggest that the Carmel area was submerged under marine conditions before the upper Miocene. They base this deduction on a single outcrop located in Bet-Rosh that contains a continuous marine succession from the Eocene to mid-Miocene. These authors accept the possibility that the Galilee was exposed and claim that it resembled the Carmel in the timing of initiation of vertical displacements during the upper Miocene. The integrative geological-geophysical data presented here show differently. Our results attest to hundreds of meters thick lacustrine-fluvial infill that accumulated during the early and mid-Miocene displacements, while tectonics were active (Figs. 7-12).

In summary, the pattern of subsidence of separated and localized basins continues from stage one to stage two. However, during the second stage, the basins also elongated along an NNE trend, while keeping the elevated structural highs in between (Fig. 12). Numerical modelling of deeper sections of the lithosphere predicted such relief pattern, of rift axis perpendicular faulting (Lyahovsky et al., 2012; Segev et al., 2014). The subsidence extended beyond the area studied here into the regions that were uplifted and eroded during the Plio-Pleistocene, for example, over H3 and the tilted blocks of the eastern Galilee (Figs. 3, 6, 7).

## 6.2 Tectonic classification of the basins during the two stages

The Galilee basins developed during the Neogene due to two major structural processes. Extensional regime during the first stage (20-9 Ma) formed the Galilee basins. The thinning of the Lower Basalt Fm. to the northwest (Figs. 6, 10) supports diminish of volcanic sources as well as shallowing of the basin floor in that direction. The Lower Basalt does not cross H1 (Fig. 3) to the west. This trend suggests a reduction in regional extension towards the continental margin in the west, previously

assessed by Freund (1970). At this stage, the structure of the basins and their dimensions are equivalent to intraplate grabens and half-graben basins that form during intra-continental rifting (Evison, 1959; Bosworth, 1994; Busby and Ingersoll, 1995; Allen and Allen, 2005; Morley et al., 2004). Previous studies showed the development of the Irbid (also referred to as Qishon-Sirhan or Azraq-Sirhan) rift during the Oligocene-Miocene in a northwesterly direction (Shaliv, 1991; Schattner et al., 2006a;

Segev et al., 2014).

The second stage of subsidence (9-5 Ma) marks a transition of the extensional stress regime into transtension along a primary NNE direction and a secondary WNW direction. Basins subsided vertically and extended perpendicularly to the principal axis of the first stage basins, while uplifted blocks (i.e. structural highs) separate between them in a NNE direction (Fig. 12). The highs are accommodation zones, structurally equivalent to the intervening block separators between basins along the East

African Rift (Bosworth, 1985; Bosworth et al., 1986; Rosendahl, 1987; Ebinger et al., 1987; Burgess et al., 1988; Ebinger et al., 1989; Morley et al., 1990). The latter studies also show that basins along a forming rift accumulate sediments while tectonic subsidence is in action. As a system, some of these rifts may succeed and continue to open, while others fail. The two stages recorded here occurred alongside the initiation of motion along the nearby DSF plate boundary.

Interaction between the Dead Sea fault and the Irbid rift is depicted by the deep depocenter of Bet Shean basin (B7) at the then

junction area (Fig. 7; pre-lateral displacement on the Dead Sea fault). Volcanism initiation is also suggested as 17 Ma for the Galilee (Rozenbaum et al., 2016; Shaliv et al., 1991). Transform-rift interaction adjacent to continental margin is manifested by NW-striking faults within the Galilee and NE-striking faults within the Golan Heights (location: Fig. 3). This process signifies the crossing of the Irbid rift into the other side of the DSF (Fig. 1b; Segev et al., 2014). Our study supports the numerical modelling of Segev et al. (2014) by showing that the active rifting of the Irbid rift on the western side of the DSF

succeeded in opening basins by cutting across the Levant continental margin (Fig. 1b).

### 6.3 Structural stress field transitions along the plate boundary

The on-going Afro-Arabian and Eurasian convergence (Letouzey, J. and Tremolieres, 1980) induced three major stress regimes across the Galilee. (1) The Syrian Arc compressional stress regime (Krenkel, 1924) produced a WNW shortening during the Turonian (Eyal, 1996; Eyal et al., 2001). Compression-related folds plunge north towards the Carmel-Gilboa trajectory, are

buried in the subsurface of the southern Galilee basins, and are exposed again across the northern Galilee (Fig. 3). (2) The Red Sea extensional regime (N60°E extension) prevailed during the Oligocene to the early Miocene (Steckler and tenBrinck, 1986; Khalil and McClay, 2002; Younes and McClay, 2002; Bosworth et al., 2005; Khalil and McClay, 2016). It resulted in the coeval opening of the parallel Red Sea and Irbid rifts (Shaliv et al., 1991; Schattner et al., 2006a). The N60°E extension (McClay and Khalil, 1998; Younes and McClay, 2002; Bosworth et al., 2005) later shifted during the Neogene (Garfunkel and

Bartov, 1977) to the NNE (N15°E, Bosworth et al., 2005). The NW trending faults developed across the study area are part of larger fault systems extending across the western Arabian plate (Fig. 1). Fault systems of the Suez-Red Sea (Steckler and ten Brink, 1986), Irbid (Schattner et al., 2006a) and Karak (Bender, 1974) reactivated traces of the Precambrian Najd fault system (Stern, 1985; Agar, 1987; Stern, 1994; Fig. 1). Our data show that the Red Sea regime provided sufficient conditions for the

first stage of subsidence of the southern Galilee basins, at the northwestern tip of the Irbid rift. The failure of this rift during the early-to-mid Miocene is closely associated with the emergence of the third, Dead Sea, stress regime (Schattner et al., 2006b; Segev et al., 2014).

Convergence between the Arabian and Eurasian plates transformed into collision and slowed down during the mid-Miocene
(14-12 Ma). This short recess resulted in tectonic quiescence in the Suez portion of the Red Sea rift (Bayer et al., 1989), the southern equivalent of the Galilee basins. In between the two rift systems (i.e. Suez and Galilee), the Negev (southern Israel), ceased to subside (Zilberman and Calvo, 2013; location: Fig. 1); while the Judea region was elevated by 400 m above the Miocene coastline (Sneh and Buchbinder, 1984; Bar, 2013; location: Fig. 1). During the same time window, a numerical simulation shows a depression that subsided along the Irbid rift NW-trending axis, still not entirely affected by the displacement
along the intersecting DSF (Lyakhovsky et al., 2012; Segev et al., 2014). This depression extended from Irbid structural low in the east (NW Jordan) to Beteha-Sea of Galilee-Kinnarot basins in the west (Location: Fig. 1B; Segev et al., 2014)..

The tectonic transition between the Red Sea and Dead Sea stress regimes was accompanied by up to 50% decrease in the relative velocity of the African plate around 11 Ma (Reilinger and McClusky, 2011), and a geometric rearrangement of the plates around 9 Ma (McQuarrie et al., 2003; Faccenna et al., 2013). This transition corresponds with the first to second
subsidence stage shift of the southern Galilee basins (Fig. 12). The DSF cuts through all previous structures along its ~1000 km trajectory. These include the Irbid rift. As a result, the southern Galilee basins, isolated from their original system, continued to extend along an orientation tangential to the new stresses. This extension appears as the second stage of subsidence of the southern Galilee basins (Fig. 12).

Previous studies widely agree on an N-S extension of the Galilee during the upper Miocene. Schulman (1962) and Horowitz
(1979) suggest that the Galilee basins continued to extend during the late Miocene. Freund (1970) calculated the finite N-S extension based on exposed faults in the Galilee. His results indicate an increase from 0% along the Mediterranean coast, through 5% across the central Galilee, and 7% in B7 (Bet Shean basin) near the DSF. This distribution pattern of displacement also corresponds to the exposure of Lower Basalt Fm. that decreases westwards. Freund (1970) related the differential N-S extension to the displacement along the nearby DSF. Ron and Eyal (1985) suggest that during the Miocene to early Pliocene
an N-S extension with E-W compression prevailed across the Galilee. These stresses resulted in lateral shear along conjugate faults, accompanied by block rotation. The NNE trending extensional basins defined in our results are in line with these deductions. The separation between first (17-9 Ma) and second (9-5 Ma) stages suggested here for the first time explains the structural relations between the declining Irbid rift and the emergence of DSF dominance. The NE extension of the Galilee during the declining rifting decreases in the second stage and shifts to NNE. However, NNE extension, including an E-W
compression component, prevails into the Pliocene (Figs. 12, 13). Plio-Pleistocene geodynamic analysis poses the study area as a seismogenic branch off the DSF plate boundary. The Primary Deformation Zone (PDZ) is expressed by a northwest oriented cross-cutting shear that overcomes basin subsidence. Earthquake epicenter distribution and mechanisms, GPS measurements and regional studies point to a seismogenic zone located at 9-17 kilometres beneath the surface (Eyal and Reches, 1983; Ron and Eyal, 1985; Ben-Avraham and Ginzburg, 1990; Eyal, 1996; Hofstetter et al., 1996; Hardy et al., 2010;

Salamon et al., 2006; Gomez et al., 2007; Shamir, 2007; Marco, 2007; Sadeh et al., 2012; Palano et al., 2013). Our tectonic analysis of the Galilean sheared margins in the frame of the Dead Sea fault localization process will be published in a separate paper (Wald, 2016).

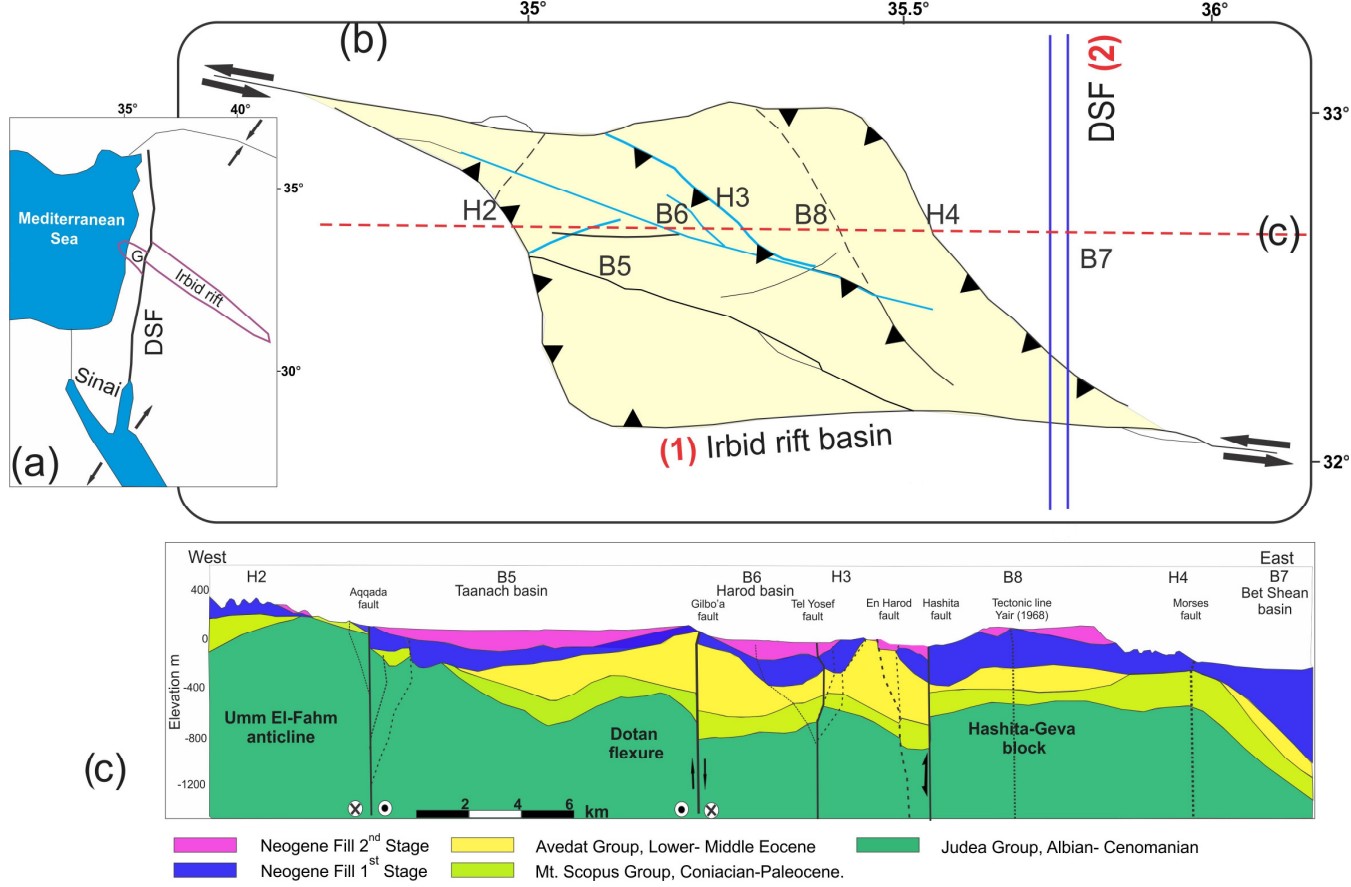

Figure 13. (a)  Current plan view of the northwest trending Irbid rift dissected by the Dead Sea fault plate boundary. (b) Neogene basin subsidence across the Galilee during the Irbid rifting (marked as 1st stage). NNE elongation provoked extension across the interpreted normal faults (marked by celeste lines). The 2nd stage reflects the Dead Sea Fault (DSF) stress regime, during which subsidence, normal faulting and graben formation decrease while complex strike-slip faulting characterizes the strain style. An establishment of a left-lateral strike-slip Primary Deformation Zone- PDZ, modified after McClay and Bonora, 2001. (c) East-west geological cross section through basins B5, B6, B8, B7 (location: Figs. 3 and 13B). The profile is extracted from a structural model constructed for the entire study area based on seismic data, wells, and outcrops. Blue and pink shading represent the 1st and 2nd subsidence stages respectively. An inverted relief of tilted blocks is a result of a Pliocene-Pleistocene ESE compressional stress component of the Dead Sea Fault stress regime. Dotted lines: less verified fault planes.

## 6.4 Failed rifts and magmatism

The low extension rate (<7%) in the Galilee corresponds to similar values in other failed rifts, such as Lake Tanganyika (Morley et al., 1990; Rosendahl, 1987). The extension is also associated with dike emplacement. Dikes may focus the strain

to detachment faults (Rosenbaum et al., 2008). In Afar and Ethiopia (eastern Africa) normal faults developed during the initial stages of rifting and were abandoned 10 Ma later. Extensional stresses there have focused on a narrow region that contains faults and magmatic intrusions (Ebinger and Casey, 2001). In the Gulf of Aden, the magmatic activity was smaller. d'Acremont et al. (2005) show an abandonment of older detachment faults within the rift environment replaced by the formation of a newer,

shorter segmentation along the central axis of the rift. Rift associated magmatism therefore commences in regions distant from the rift axis, and is dependent on fault distribution. In systems where extension is localized to narrow zones, dikes may follow extension lineaments. Examples of such cases are the Gulf of California (Lizarralde et al., 2007) and along the magmatic boundary of the north Atlantic (White et al., 2008). In both areas, the basaltic intrusions appear within the narrow 50-100 km outline of the rift. Hence evidence for magmatic intrusions and their spatial arrangement may hint at rifting orientation and

associated extensional stresses.

During the first stage of subsidence in the Galilee volcanism arrived mainly through extensional lineaments associated with normal faulting, along with the subsidence of the basins (Fig. 12). Syn-tectonic volcanism supplied the thick sections of Lower Basalt and Hordos Fm. in B4, B6, and B7 (Figs. 6, 9, 10b). A volcano in the southern margin of B6 and possible sources along H2 supplied additional volcanics that accumulated in B2, B4, and B5. The magmatic intrusions in H3 (Givat Hamore: location:

Figs. 3, 10, 7) were dated to 15 Ma and associated with an NW to WNW faulting system (Fig. 7; Dicker, 1964; Shaliv, 1991). Volcanism continued during the second stage of subsidence, along with the vertical and horizontal displacement of the study area. The Intermediate Basalt Fm. dated to ~6 Ma arrives through normal faults bounding H3 from the NE, and perhaps through a volcano located in the Rewaya block (Shaliv, 1991; Fig. S8). The directional correlation between faulting and volcanic centers and lineaments (Figs. 7, 10, 12) obeys to a similar regional tendency. Equivalent correlation appears in Karak graben

(Bender, 1974), Miocene dikes across Sinai (Bartov et al., 1980; Baldridge et al., 1991), and across Harrat-A-Shaam volcanic field (Feraud et al., 1985; Mor, 1986; Giannérini et al., 1988; Brew et al., 2001; Al Kwatli et al., 2012). The strips of alkaline volcanism across the Arabian plate represent the beginning of Miocene volcanism (Camp and Roobol, 1992; Weinstein, 2000; Ilani et al., 2001). We, therefore, suggest that the faulting and volcanism of the southern Galilee also follow weak lineaments in the lithosphere.

The timing of regional volcanism is noteworthy. Between 18-12 Ma volcanic activity ceased across the Arabian plate and was dominant across the southern Galilee basins (Lower Basalt Fm.). This shift may represent an NW propagation of extension and volcanism across the Arabian plate (Weinstein, 2000). The northwestern Arabia volcanism was renewed at 14-12 Ma (Bohannon et al., 1989; Camp and Robool 1992; Ilani et al., 2001; Krienitz et al., 2009). Several studies link the renewal and activity with structural aspects (Bayer et al., 1988; Camp and Roobol, 1992; Ebinger and Casey, 2001). However, other studies

suggest that the lateral slip along the DSF decreased during the upper Miocene (Hempton, 1987; Bayer et al., 1989; Reilinger and McClusky, 2011; Faccenna et al., 2013), while drift across the NW trending Irbid rift was active (Segev et al., 2014; Segev et al., 2017). Our results suggest that this decrease also enabled the subsidence of the southern Galilee basins during the second stage, as part of the hybrid Red Sea - Dead Sea stress regime. With enhancement of motion along the DSF during the early

Pliocene around 5 Ma, the Dead Sea stress regime became dominant, laterally shifting the southern Galilee basins, and structurally isolating them from their first association to the Irbid rift.

## 7. Conclusions

The Galilee basins subsided along the northwestern front of the Irbid rift. Integration of geological and geophysical data bounds the subsidence of the basins between two major surfaces: the Oligocene Regional Truncation Surface (RTS) and the top of Bira Fm. unconformity. The subsidence is divided into two stages.

During the first stage (20-9 Ma) the Galilee basins subside along the main trend of the Oligo-Miocene Irbid rift system. They subside as grabens and half-grabens, bounded by normal faults and structural saddles. Larger subsidence was recorded along the main NW trending rift axis. Smaller basins subsided off the main axis. The subsidence occurred along with extensive volcanism that arrived through fault planes that bound the basins. The spatial arrangement of the rift basins suggests that they follow a larger Principal Displacement Zone (PDZ). The major boundary faults mapped here are the surface expression of the PDZ strands that bound the basin complex of the rift from north and south. The complex originally formed as a releasing jog along a rift system. The structural change around 9 Ma is associated here with the gradual transition between the Red Sea and the Dead Sea stress regimes. With the initiation of shearing along the DSF, the jog and its basins were truncated. The transition elongated the basins, accentuated their subsidence, and uplifted their surrounding margins.

During the second stage (9-5 Ma) left lateral shearing of the entire study area results in subsidence of a series of NNE trending basins, perpendicular to the major axis of basins from the first stage. Structural highs (i.e. blocks) that divide between the first-stage basins remained high during the second stage. However, during the second stage, their bounding normal faults also exhibit a lateral component. The shear distorts the original structure of the first stage basins north and south of the major NW-trending axis, in a manner that today these periphery early Neogene basins have been uplifted and weathered. The length of the basins decreases from ~60 km in the east to ~15 km in the west of the study area. The volcanism of the second stage arrives from weak zones and focusses on structural boundaries between the basins, and volcanic activity along their margins.

Structural architecture of the southern Galilee indicates that the rift basins continued to subside while the Irbid rift was active. Their shape and arrangement were constrained by two main rheological features – the bounds of a releasing jog along the PDZ (i.e. Carmel-Gilboa fault line) and the acquaintance with a more cohesive crust at the peripheral area, perhaps a "locked zone" (see Lyakhovsky et al., 2012; Segev et al., 2014). Locked zones involve pre-existing discontinuities suchlike transition between oceanic and continental crust types or perpendicular faulting arrays (Courtillot et al., 1987; Dunbar and Sawyer, 1996). However, following the numerical modelling results, neither of these seems to have caused the cessation of rifting. In fact, the basins at the rift tip subsided until the jog was decapitated by the motion along the DSF. Following the two-staged subsidence model of the Oligo-Miocene, the main cause of the structural transition (and preservation) of the southern Galilee basins was the transition from one dominant stress regime to another. Our study provides a unique and detailed architecture of a rift

termination basin complex. Based on this case study we suggest that the rift front did not fail but rather faded and was taken over by a more dominant stress regime. Otherwise, basins of this failing rift front could have simply died out.

**Author contribution**

This work is based on a profound chapter from Reli Wald's PhD thesis. Reli Wald has processed and analysed the datasets, including seismic interpretation and development of a 3D geological model. Amit Segev, Zvi Ben-Avraham and Uri Schattner have critically read and reviewed all the data following their participation in work as thesis advisors. Uri Schattner has contributed in writing and in figure graphics. Reli Wald has prepared the manuscript with major contribution from Uri Schattner and with review of co-authors.

**Acknowledgements**

We thank both the Israel Geological Survey (Jerusalem) and the Graduate Studies Authority of the University of Haifa for funding this research. We are grateful to John Hall, Anat Kedem, Devrim Tezcan for their help in processing seismic datasets. We thank the Kingdom Suite and Schlumberger-Petrel for providing academic licenses that facilitated this study. The Kingdom Suite software has been used for archiving and editing the data whereas Schlumberger-Petrel has been used for 3D-modelling and geologic 2D slices presentations. This manuscript highly benefited from the productive reviews of Piotr Krzywiec and an anonymous reviewer.

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
