# Peer review of "Structural expression of a fading rift front, a case study from the Oligo-Miocene Irbid rift of northwest Arabia"

_Solid Earth, 2018_

## Referee Comment (RC1) · Anonymous Referee #1 · 10 Oct 2018

The paper entitled "Structural expression of a fading rift front, a case study from the Oligo-Miocene Irbid rift of northwest Arabia" is a very good contribution to the geology of the Levant. The manuscript is well organized and very well written. Some minor corrections/clarifications need to be done: 1) In Figure 1, please add geographic coordinates. This recommendation is also valid for other maps that do not present any coordinate system, such as in figures 12 and 13 2) Still in Figure 1, the word Judea is presented but the references for Judea in the text are for a formation or a group. In this sense, what is the correct meaning for Judea in the Figure and, is Judea a formation or a group? 3) Actually, the second stage tectonic reactivation seems to last until the beginning of the Pliocene (is it?) In this case, should the authors change the title of the

paper? Also, despite it is not the aim of the paper, it would be good to read one or two sentences about "the end of the story", i.e. the Plio-Pleistocene evolution.

I recommend its publication after these minor corrections

———————————————

---

## Author Comment (AC1) · 12 Oct 2018

We are very thankful for this productive interactive comment of anonymous referee #1. Indeed an important aspect of this manuscript is the sequence of events depicted from the sedimentary and facies architecture. Hence, we are content that we managed to deliver the story clearly.

In regard with the above corrections/clarifications we did the following: (1) We added coordinates to figures 1, 12, 13. We would like to mention that figure 13b is based on a numerical model of McClay and Bonora (2001) that according to our analysis fits the Galilee (northern Israel, Irbid rift northwestern front) case. (2) Figure 1, "Judea":

[Figure]

The Judea label is neither a group nor formation in Figure 1. It denotes a geographical region where the Judea Group type section was defined. In order to clarify this issue, we modified the label to "Judean Hills". (3) True. The second stage of basin subsidence ends at roughly 5 Ma, while the Dead Sea stress field persists into the Plio-Pliocene. Evidence for activity on segments of the Carmel-Gilboa fault system, off the plate boundary is dealt with in the PhD thesis and in a future publication. However, we think that modification of the manuscript title will not serve its contribution to the scientific community. Changing the current "Oligo-Miocene" to "Cenozoic" for example, is too vague and does not focus on the evolutionary stage of the rifting.

The ending paragraphs of the manuscript include a new "closure", to clarify that segments of the fading rift still show life signs in terms of earthquakes and vertical offsets. These last gasps of activity are shown in figure 13 by reversal of fault movement sense and uplifted high blocks that remain high as yet. The youngest evolution stages of the study area are firstly mentioned in the introduction, Chapter 1.1, lines 5-11. In the Results chapter it is dealt with in lines 19-21 of chapter 3.2.1 and shown by vertical offsets of late Miocene-Pliocene units in Figure 11. Finally, in the Discussion chapter the Plio-Pleistocene evolution is further exemplified in a larger geodynamic frame. It is presented by lines 7-26 in chapter 4.1.2, that lead the reader to understand that after the relief has been filled by sediments of the second stage of basin subsidence, the basins had been cut through by Plio-Pleistocene shear, part of which prevails to date. Structural relations between the declining Irbid rift and the emergence of Dead Sea fault dominance are shown in Figures 12, 13 and leave the reader with a sense of closure. Following Referee #1's advice, we added a few sentences at the end of chapter 4.4, lines 21-27. However, since we intend to focus on the Plio-Pleistocene shear in a future publication, we only wrap-up the story by hinting to it in the closing sentences of the Conclusion chapter.

Kind regards, Reli Wald on behalf of the authors

Please also note the supplement to this comment:
https://www.solid-earth-discuss.net/se-2018-91/se-2018-91-AC1-supplement.pdf

[Figure]

**Supplement:**

[revised manuscript text omitted]

Qishon-Sirhan rift during the Oligocene-Miocene in a northwesterly direction (Shaliv, 1991; Schattner et al., 2006a; Segev et al., 2014). Results of the present study claim this rift comprises the southern Galilee Basins during their first stage of development.

The second stage of subsidence (9-5 Ma) marks a transition of the extensional stress regime into transtension along a primary NNE direction and a secondary WNW direction. Basins subside vertically and extend perpendicularly to the principal axis of the first stage basins, while structural highs separate between them (Fig. 12). The highs are accommodation zones, structurally equivalent to the separators between basins along the East African Rift (Bosworth, 1985; Bosworth et al., 1986; Rosendahl, 1987; Ebinger et al., 1987; Burgess et al., 1988; Ebinger et al., 1989; Morley et al., 1990). These studies also show that basins along a forming rift accumulate sediments while tectonic subsidence is in action. As a system, some of these rifts may succeed and continue to drift, while others fail. The two stages recorded here occurred alongside the initiation of motion along the nearby DSF plate boundary. While motion took place along the entire boundary between the Arabian and Sinai plates, intracontinental basins subsided only across the southern Galilee.

[revised manuscript text omitted]

---

## Author Comment (AC2) · 30 Nov 2018

We are very thankful for this thorough and productive interactive comment of referee #2, P. Krzywiec. This review significantly improved the paper in fine tuning of the evidence. Not only technical issues suchlike vertical exaggeration and fault imaging were dealt with, but also careful examination of our evidence from the rift front (i.e. termination) west of the Dead Sea fault in relation with the main body of the rift east of the Dead Sea fault.

We list below the main subjects attended in this thorough review.

[Figure]

All of the detailed corrections according to the referee's comments are submitted in the attached pdf file. We will follow-up in another reply with a clean revised manuscript for the reader's convenience.

The whole Irbid rift in comparison to its western front: The term "rift front" refers to the rift zone termination sensu Lyakhovsky et al. (2012) and Segev et al (2014), that is the terrain in which the rifting process decays. A nearby example is northern Sinai referring to the Suez rift termination (Steckler and ten Brink, 1986; Segev et al., 2017). An explanation to this term has been added in the abstract "Our results show that a series of basins subsided at the rift front, i.e. rift termination, across the southern Galilee". Figure 1 has been updated, (a) to include a schematic extent of the Irbid rift, (b) to present the full structure of the Irbid rift across both sides of the Dead Sea fault.It should be noted that the study area is located off the Dead Sea transform, a major plate boundary. Plio-Pliocene uplift effected the plate boundary shoulders. A wide damage zone of at least 50 km from each side of the plate boundary has been affected (Wdowinski and Zilberman, 1997). More specifically, the interaction between the Irbid rift and the lateral offset on the Dead Sea fault has been stressed out in the discussion. In the Galilee, basin subsidence sheltered the data at relatively shallow depths of up to 1 km. For depth cross sections of the Irbid rift vicinity see also figure 6 in Segev et al. (2014), a geologic cross-section from SW Jordan to NE Syria across the Azraq-Sirhan graben (modified after Konert et al., 2001 and references therein) and Luening and Kuss (2014) In: Petroleum Systems of the Tethyan Region, AAPG, Editors: Marlow, L, Kendall, C and Yose, L.A.

Faulting characterization: The referee has commented about the lack of clear vertical subsidence of the Cretaceous units (Judea and Mt. Scopus Groups) at the bottom of the seismic sections (Fig. 5 and 8). In the Galilee, most imaged faults began as normal, but later deformation obscures the reflectors near the fault planes and fault branches in addition to lower resolution above 0.5 msec two way time. Actually, the Galilee amalgamates significant geodynamic stages discussed in the paper. Seismic profiles

reflect the current structure hence need careful observation. Multi-stage deformation causes slip reversals on fault planes, folding, branching and other phenomena that inhibit clear imaging of the Neogene tectonics. Even so, the vast amount of data, counting in the supplementary files attached to this manuscript provides a sound base supported by intersecting seismic profiles and well data. In addition, isopach maps and structural maps further support normal faulting. The figure captions have been rephrased to further verify the fault types. For example, the Nahalal fault shown in Figure 11, is currently a strike-slip fault with a very distinct flower structure also verified by cross lines and from EQ focal solution. It did begin as a normal fault down-faulting Kefar Baruch basin to the south. Kefar Baruch and Gevat faults are normal but have a strike-slip sense of movement that might explain the folding of Neogene layers adjacent to the fault planes. The graben between these two faults is verified by potential methods (Segev and Rybakov, 2011) and by crossing seismic lines interpreted herein. We focus on the Pliocene-Pleistocene faulting characterization in a future publication.

Irbid rift /Azraq-Sirhan rift/ Qishon-Sirhan rift: All of the names above define the same rifting episode. Azraq-Sirhan has been used in Shaliv (1991) to asses a Neogene internal continental basin. Qishon-Sirhan has been used in Schattner's publications in 2006, to assess the northwest trending rifting from northern Jordan (Azraq-Sirhan) to Haifa Bay, later offset by the Dead Sea fault (Qishon river is the main catchment of the southwestern Galilee area). Here we use the Irbid rift following Segev et al. (2014) and Segev et al. (2017) recent publications, to further establish the Neogene geodynamic evolvement of the study area, at the northwest termination of the Irbid rift (Irbid rift front).

Locations, and local nomenclature: Our study area stretches across a very narrow terrain, immediately to the west of the Dead Sea fault plate boundary. However it plays an important role in the development of the NW oriented Oligo-Miocene rifting. Basin subsidence and more importantly subsurface preservation provide an important architecture, including depocenters and fault arrays that are very relevant to future EQ

assessments. Several focal solution surveys have already been done to model this seismogenic splay off the plate boundary (Hofstetter et al., 1996; Shamir, 2007). The largest EQ of the past decades was that of magnitude 5.4, 1984- with the epicenter in the midst of our basins. This study provides a detailed, high resolution structure never explored before. For this reason, we believe that most of the names, all of which are supported by figures and supplementary files, should stay as is in the manuscript. The names have been classified to basin codes (e.g. B1, B2...B11) and uplifted blocks/highs codes (H1-H4) to avoid extra names. Our main base for the model are small-scale basins and their border faults. That is why we separate the presentation of them in the Results chapter. Separation and hierarchy (e.g. major vs secondary faults) allows relatively short chapters that provide the most important issues in regard with basin subsidence. We put effort in providing short and clear opening sentences at the beginning of each paragraph. That way, readers from abroad could interact with the data if they do not want to dive into minor details. Our presentation of subjects is logic (e.g. geographically oriented, from the big picture to the small details).

Vertical exaggeration (referring to figures 5, 8, 11 in the manuscript): A common issue found in published papers presenting seismic reflection data. Most of the times exporting the profiles from the professional interpretation software already creates a hidden exaggeration not accounted for. An interesting publication regarding this phenomena states that approximately 75% of the seismic data published involves at least x5 vertical exaggeration! (Stewart, 2011). We followed the referee's advice to stretch the seismic section (x1.5 horizontal stretch, x0.8 vertical squeeze). We believe that the new dimensions allow better presentation of the data. Stewart, S.A.: Vertical exaggeration of reflection seismic data in geoscience publications. Marine and Petroleum Geology, 28, 959-965, doi:10.1016/j.marpetgeo.2010.10.003, 2011.

Supplementary material: This study relies on vast geological and geophysical information, part of which is included in the Supplementary folder to support our results and discussions. Our supplementary files (including the Google Earth file) are very

significant for the understanding of the area. Part of our responds to the referee's comments are found in these datasets.

Please also note the supplement to this comment:
https://www.solid-earth-discuss.net/se-2018-91/se-2018-91-AC2-supplement.pdf
* * *
[Figure 1: Tectonic map of the Eastern Mediterranean and Arabian region]

**Fig. 1.** Figure 1A- minor additions

[Figure]

**Fig. 2.** Figure 1B new, modified after Segev et al., 2014

[Figure]

**Fig. 3.** Figure 5- corrected

[Figure]

**Fig. 4.** Figure 8- corrected

[Figure]

**Fig. 5.** Figure 11- corrected

**Supplement:**

[revised manuscript text omitted]
. (2006), Nelson et al. (2009), Zilberman et al. (2009). Structural maps constructed from interpreted key surfaces include associated faults. The surfaces and faults were exported from the Kingdom Suite to Petrel (Schlumberger) to build a structural model. Previous geological mapping assumed to extend the structural model from sea level datum (elevation of 0m) up to the present-day topography (30-550m asl). Using a 2000 m/sec velocity for the shallow, near-surface beds (weathered beds), enabled

25    correlation between depth and time domains. Completion of the structural model relied upon digitization of location surfaces from previous studies in ArcMap (ESRI) (Weiler, 1968; Dicker, 1969; Dekel, 1988; Shaliv, 1991; Shaliv, 2003; Sneh, 2008). Outcropping truncated surfaces are considered as layers within a specific unit rather than its top (due to erosion). Control points were added from boreholes. Integration of all datasets yielded a coherent database and a three-dimensional geological grid model of the Galilee subsurface, extending from a depth of 2500m to the present-day surface topography.

**3. Results**

[Figure]

The results section describes the sedimentary fill of the basins in chronological order. It is followed by a description of the structural elements. Local names of the basins and the structural highs are abbreviated to simplify the description (Fig. 3). The geographical location of all sites mentioned in text appear in the Google Earth™ supplementary material, herein referred to as GE. The sedimentary fill is bounded between two temporal and structural markers. The basin floor is marked by the Oligo-Miocene Regional Truncation Surface (RTS; ~23-17 Ma; Figs. 2, 4), a peneplain predating the subsidence of the basins. The RTS truncates the folded and displaced structures of the Judea, Mt. Scopus and Avedat Groups (Fig. 2). The latter is  towards H2 and pinches out approximately 400 meters west of it (Fig. 5). The imentary fill culminates upwards up to the top of Bira Fm. that formed a very low relief. The Cover Basalt Fm. locally covers the Bira Fm., providing a temporal marker. alysis 
[revised manuscript text omitted]

hon-Sirhan rift during the Oligocene-Miocene in a northwesterly direction (Shaliv, 1991; Schattner et al., 2006a; Segev et 2014). Results of the present study claim this rift comprises the southern Galilee Basins during their first stage of development.

The second stage of subsidence (9-5 Ma) marks a transition of the extensional stress regime into transtension along a primary NNE direction and a secondary WNW direction. Basins subsid rtically and exten rpendicularly to the principal axis of the first stage basins, le structural highs separate between them (Fig. 12). The highs are accommodation zones, structurally equivalent to the arators between basins along the East African Rift (Bosworth, 1985; Bosworth et al., 1986; Rosendahl, 1987; Ebinger et al., 1987; Burgess et al., 1988; Ebinger et al., 1989; Morley et al., 1990). se studies also show that basins along a forming rift accumulate sediments while tectonic subsidence is in action. As a system, some of these rifts may succeed continue to t, while others fail. The two stages recorded here occurred alongside the initiation of motion along the nearby DSF plate boundary. While motion took place along the entire boundary between the Arabian and Sinai plates, intracontinental basins subsided only across the southern Galilee.

[revised manuscript text omitted]

---

## Author Response (AR1)

**Referee #1 (anonymous)**

| Referee Comment | Authors answer | Revised text in manuscript |
|---|---|---|
| In Figure 1, please add geographic coordinates. This recommendation is also valid for other maps that do not present any coordinate system, such as in figures 12 and 13 | Figure 1- followed. Figure 12, 13- followed. | |
| Still in Figure 1, the word Judea is presented but the references for Judea in the text are for a formation or a group. In this sense, what is the correct meaning for Judea in the Figure and, is Judea a formation or a group? | The "Judea" label in Figure 1 is neither a group nor formation. It denotes a geographical region where the Judea Group type section was defined. To clarify this issue, we modified the label to "Judean Hills". | |
| The second stage tectonic reactivation seems to last until the beginning of the Pliocene (is it?) In this case, should the authors change the title of the paper? | True. The second stage of basin subsidence ends at roughly 5 Ma, while the Dead Sea stress field persists into the Plio-Pliocene. Evidence for activity on segments of the Carmel-Gilboa fault system, off the plate boundary is dealt with in the PhD thesis and in a future publication. However, we think that modification of the manuscript title will not serve its contribution to the scientific community. Changing the current "Oligo-Miocene" to "Cenozoic" for example, is too vague and does not focus on the evolutionary stage of the rifting. | |
| Despite it is not the aim of the paper, it would be good to read one or two | The ending paragraphs of the manuscript include a new "closure", to clarify that segments of the fading rift still show life signs in terms of earthquakes and vertical offsets. These last gasps of activity are shown in figure 13 by reversal of fault movement sense and uplifted high blocks that remain | **End of chapter 4.3 (later 6.3): Structural transitions along the plate boundary:** Plio-Pleistocene geodynamic analysis poses the study area as a seismogenic branch off the |

| | | |
|---|---|---|
| sentences about "the end of the story", i.e. the Plio-Pleistocene evolution. | high as yet. The youngest evolution stages of the study area are firstly mentioned in the introduction, Chapter 1.1, lines 5-11. In the Results chapter it is dealt with in lines 19-21 of chapter 3.2.1 and shown by vertical offsets of late Miocene-Pliocene units in Figure 11. Finally, in the Discussion chapter the Plio-Pleistocene evolution is further exemplified in a larger geodynamic frame. It is presented by lines 7-26 in chapter 4.1.2, that lead the reader to understand that after the relief has been filled by sediments of the second stage of basin subsidence, the basins had been cut through by Plio-Pleistocene shear, part of which prevails to date. Structural relations between the declining Irbid rift and the emergence of Dead Sea fault dominance are shown in Figures 12, 13 and leave the reader with a sense of closure. Following Referee #1's advice, we added a few sentences at the end of chapter 4.3, lines 21-27. However, since we intend to focus on the Plio-Pleistocene shear in a future publication, we only wrap-up the story by hinting to it in the closing sentences of the Conclusion chapter. | DSF plate boundary. The Primary Deformation Zone (PDZ) is expressed by a northwest oriented cross-cutting shear that overcomes basin subsidence. Earthquake epicenter distribution and mechanisms, GPS measurements and regional studies point to a seismogenic zone located at 9-17 kilometres beneath the surface (Eyal and Reches, 1983; Ron and Eyal, 1985; Ben-25 Avraham and Ginzburg, 1990; Eyal, 1996; Hofstetter et al., 1996; Hardy et al., 2010; Salamon et al., 2006; Gomez et al., 2007; Shamir, 2007; Marco, 2007; Sadeh et al., 2012; Palano et al., 2013). Our tectonic analysis of the Galilean sheared margins in the frame of the Dead Sea fault localization process will be published in a separate paper (Wald, 2016). |

**Referee # 2 Piotr K, chronologically ordered according to the supplement pdf file submitted on November 16, 2018.**

\* It should be noted that Referee #2 reviewed the original pdf manuscript, prior to our corrections following referee #1. Hence, our corrections after referee #1 were not seen in the pdf file of referee #2 comments. However, only one correction of referee #1 involved text revision (end of chapter 4.3 in the discussion (later revised to 6.3), i.e. the paragraph preceding figure 13. Other corrections following referee #1 are merely figure corrections and self-corrections of 6 missing references that were shown in text and absent from the list.

| Referee Comment | Authors answer | Revised text in manuscript |
|---|---|---|
| 1.PIRATED: what exactly do you mean by this, what kind of geological process ...? 2. So this "pirating", whatever this exactly means, was the sole reason for rift decease? | 1. Pirated changed to "taken over". 2. Rephrased to: ...and mostly due to that… | The current study focuses on a uniquely preserved Oligo-Miocene rift that was subsequently *taken over* by a crossing transform fault system and *mostly due to that* died out. |
| 1. "…unravel the structural development of the Irbid failing rift…" Replace: NW segment of the Irbid rift 2. "Despite tectonic, magmatic and geomorphologic activity postdating the rifting, its subsurface structure is preserved at depths of up to 1 km". Of the entire Irbid rift zone or only its NW segment located NW from the DFZ? 3. What is "rift front"? 4."The basins continued to subside until a transition from | We revised following the referee's comments. Revisions are shown in italics in the next column. 1. Accepted. 2. Geographical specification of the study area. 3. We added an explanation after the term. 4. We placed adjectives before the stress field names (reported and discussed in literature since the late 1980's) to verify the local significance. | 1. We integrate all geological, geophysical and results from previous studies from across the Southern Galilee to unravel the structural development of the *NW segment of the Irbid rift*, northwest Arabia. 2. Despite tectonic, magmatic and geomorphologic activity postdating the rifting, its subsurface structure *northwest of the Dead Sea Fault*...is preserved at depths of up to 1 km. 3. Our results show that a series of basins subsided at the rift front, *i.e. rift termination*, across the southern Galilee. 4. "...until a transition from the *transtenssional* Red Sea to the *transpressional* Dead Sea stress regime occurred". |

| | | |
|---|---|---|
| the Red Sea to Dead Sea stress regime occurred." I understand what do you mean here but terms "Red Sea stress regime" and the "Dead Sea stress regime" might not be 100% clear for readers of this paper. 5. "… a structural as well as tectonic context to the southern Galilee basins". | | 5. "… a structural as well as tectonic context _for_ the southern Galilee basins". |

**Introduction**

| | | |
|---|---|---|
| A rapid stop  extensional stress decay, | | A rapid stop _might be a result_ of extensional stress decay, |
| Fading  of the dominant rifting stress leads to attenuation and eventually rift  abortion. | Accepted and revised. | Fading, _i.e. gradual decrease_ of the dominant…and eventually rift _abortion_. |
| "…Neocomian-Barremian syn-rift grabenization style". I'm not sure what exactly this mans ... | This term has been used in several publications of our research group, meaning graben formation. Corrected to the complete form. | "Neocomian-Barremian syn-rift _graben formation_ style". |
| 1."Magnetic, gravity and resistivity data track transform boundaries , generating fault-controlled depressions. 2."In southeastern Australia,  transform fracture zone cuts across preexisting basement structures". | 1. Revised according to suggestion.

2. Followed, strike-through term replaced by the word "a": "In southeastern Australia, _a_ transform fracture... | 1."Magnetic, gravity and resistivity data _delineated intraplate_ transform boundaries _which generated_ fault-controlled depressions."

2."In southeastern Australia, _a_ transform fracture zone cuts across preexisting basement structures". |

| | | |
|---|---|---|
| 1."In some cases, the internal architecture of the basins comprising a failed rift may be lost due to tectonic inversion, severe erosion or even burial into greater depths..."

 2. This I do not understand - how burial alone might result in destruction of rift internal architecture? Maybe you mean that imaging (seismic etc.) of such deeply buried rift zones is very difficult?

 3. "The current study focuses on the structural development of a rift front during its failure and later preservation….
 4. "… the original subsurface structure of the failed rift is preserved at depths of up to 1 km (Fig. 1)".
 -Fig. 1 does not contain a cross-section showing depth structure of this rift zone.
 . | 1. The word "some" was added before cases.

 2. Comment followed. Text revised.

 3. Rephrased according to comment: the structural development of a rift front, its failure and later preservation.

 4. Deletion accepted.
 Fig. 1B is a new figure, showing a structural map that was added in the revised manuscript, to further visualize the rift structure. See Figure 1B two rows below. | 1+2 revision:
 "In *some* cases, the internal architecture *and thus the imaging resolution* of the basins comprising a failed rift may be lost due to tectonic inversion, severe erosion or even burial into greater depths".
 3. "The current study focuses on the structural development *of a rift front, its failure and later preservation*…"

 *"We concentrate on the Irbid Rift (*also referred to as Azraq-Sirhan or Qishon-Sirhan rift*).." |
| Figure 1:
 1. Change this to rectangle - equivalent to what is shown on Fig. 3.
 2. Sinai sub-plate
 3. Figure caption:
 SG-Sea of Galilee; DS- Dead | Followed. Rectangle polygon of study area. "Sinai sub-plate" instead of "Sinai".

 3. SG- Sea of Galilee stays and is also shown in Figure 3.
 HB- Hula Basin – removed from caption. | Sinai plate has been Corrected throughout the manuscript to Sinai sub-plate. |

| | | |
|---|---|---|
| Sea; LRB- Lebanese Restraining Bend; H- Hula Basin. | | |
| **Figure 1b- a new figure.** Following referee #1 comment re: *the deeper structure of the Irbid rift.* Instead of a cross section we provide a structural map of a late Cenomanian surface encompassing the whole rift body from both sides of the modern Dead Sea fault plate boundary. | (b) Reconstructing the pre-DSF plate configuration using the structural map of the top Judea Group interface (modified from Segev et al., 2014). The Beteha basin on the Arabian plate is attached to the Bet She'an basin on the Sinai sub-plate, which showed a ~55 km motion along the DST. Basins referred to herein are marked by the B-series (B1 to B11). Abbreviations: F., fault; V., valley; B., basin; L., low; Mt, Mount; Hasb., Hasbaya; Rach., Rachaya; Serg., Serghaya; SAF, Sheikh Ali fault; BSKB, Bet She'an Kinneret Basin; DST, Dead Sea transform. | |
| **2. Regional geological setting** | | |
| Title revised to "2. Regional geological setting", from "1.1 Geological setting". | Followed. New number formatting follows the referee's advice. | |
| "Palmyride rift" where located? - all the geological units etc. mentioned in text should be shown on figures | Brackets added to verify that the palmyride belt is the present geological structure overlapping the Paleozoic palmyride rift area. | Brackets added:"... Palmyride rift *(currently Palmyride Belt in Fig. 1).."* |
| "The stresses inverted the extensional grabens…" -Of what age? | Followed. Sagy et al. (2017) reference added. | "The stresses inverted the extensional grabens *formed 100-200 m.y. earlier* and folded the Levant margin (*Sagy et al., 2017*)" |
| "… tectonic and thermal quiescence led to vertical subsidence of NW Arabia". -so what would be mechanism of this subsidence if it took place during tectonic and thermal quiescence ...? | Revised to: Gravitational vertical subsidence | "...led to *gravitational* vertical subsidence of NW Arabia" |

| | | |
|---|---|---|
| "Lower-middle Eocene sediments comprise mainly chalks and limy chalks with sporadic cherty nodules and layers."
 - layer of what? | It meant cherty nodules and layers of chert.
 Rephrased, see next column. | "… comprise mainly chalks and limy chalks with sporadic chert nodules and *chert* layers." |
| "Mantle upwelling of the Afar plume began at 34 Ma, uplifting the overlying crust. Part of the volume propagated away from the plume head northwards…"
 - show on inset on Fig. 1
 - add references
 - volume of what? | -Added to Fig. 1 inset;
 -References for the first sentence were added.
 -Mantle plume volume, corrected. | "Mantle upwelling of the Afar plume began *at the Late Eocene*, uplifting the overlying crust (*Hofmann et al., 1997; Pik et al., 2003; Avni et al., 2012 and references therein*). Part of the *mantle plume volume* propagated away from the plume head northwards…" |
| "The uplift was …"
 -The Afar uplift | Rephrased. | "The *Afar* uplift…" |
| The northwestern front of the Irbid rift crosses the southern Galilee (Fig. 1…).
 - this is actually not very clearly shown on fig. 1 | Corrected in Fig. 1. | The schematic extent of the Irbid rift has been added to figure 1, including its northwest portion. In addition, Fig. 1B (new) shows the structure of the rift from both sides of the Dead Sea Fault. |
| The Irbid rift divides  two crustal terranes that differ in thickness and seismicity, and possibly  two different sub-plates.
 - between: delete
 -consist: change to form | Corrected according to strikethroughs of the referee. | The Irbid rift *divides two crustal* terranes that differ in thickness and seismicity, and possibly *form* two different sub-plates. |
| The Galilee basins subsided during the terminal stages of Irbid rift.
 - i.e. when exactly? | Corrected. | "The Galilee basins subsided during *the late Oligocene-Miocene* terminal stages of Irbid rift. |
| -General comment before Figure 2:
 I'd suggest to add simplified regional cross-section of the | That is an important suggestion.
 Few works were done on the crust structure from both sides of the Dead Sea fault plate boundary. The most recent are those of Segev and Lyakhovsky with co-authors between | Figure 1B has been added to show the deep structure of the rift from both sides of the Dead Sea fault. |

| | | |
|---|---|---|
| Irbid rift and adjacent plats from its central part, in order to illustrate gross tectonic style of the main structure analyzed in this paper | 2011-2017. The main problem in showing such a cross section is the Pliocene to recent arching from both sides of the plate boundary,that distorts the Oligo-Miocene structure (Wdowinski and Zilberman 1997). So, we decided to **add Figure 1b** that shows the late Cretaceous structural map of the rift from both sides of the Dead Sea fault. This is a modelling of Segev et al. (2014) taking into count 55 km of lateral offset restoration. | |
| Figure 2: Explain this column. | Time in m.y. added to second column from left. | Time in m.y. |
| "… newly formed Sinai plate". -or sub-plate? - terminology should be consistent through the paper | Changed to sub-plate throughout the manuscript. | Sinai sub-plate. |
| "Transtenssion along the DSF resulted in further subsidence of basins along its  during…" -change to axis | Corrected. | Transtenssion along the DSF resulted in further subsidence of basins along its axis during |
| "However, since 5 Ma subsidence of the DSF basins accentuated (Gulf of Aqaba, Dead Sea, and Hula basins…)" - clarify why these units are listed here. | These are basins along the Dead Sea fault that document an accentuated subsidence around 5 Ma. This subsidence is also documented in the Irbid rift basins. We did revise a little the sentence and added e.g. before the basin names in brackets. | "However, since 5 Ma subsidence of *basins along the DSF* basins accentuated (*e.g.* Gulf of Aqaba, Dead Sea, and Hula basins…)" |
| Figure 3 caption: self-addition of the authors to note the highlighted boundary faults. | We highlighted these faults in Figure 3b; | We added "Note major faulted boundaries: the Carmel (C)-Gilboa (GL) southern fault boundary and the Zurim escarpment northern fault boundary". |
| **3. Morpho-tectonics of the southern Galilee basins** | | |
| Chapter number revised to 3. | Followed. | |
| "The Carmel-Gilboa and Zurim fault systems…" | Followed. We highlighted these faults in Figure 3b; | "The Carmel-Gilboa *(C, GL in Fig. 3b)* and Zurim fault *(Z in Fig. 3b)* systems…" |

| -Show on Fig. 3 | | |
|---|---|---|
| "The surface of the westernmost, Yizre'el (B2)… " | Typo, missing word: basin. Added. | "The surface of westernmost basin, Yizre'el (B2)…" |
| "To the east Kesulot (B3) and Taanach (B5) valleys are at 60-100 m, Harod valley (B6) is between…" | Replaced to basin and basins accordingly. | " To the east Kesulot (B3) and Taanach (B5) basins are at 60-100 m, Harod basin (B6) is between …" |
| The low relief of the southern Galilee basins divides between two segments of the Mesozoic Syrian arc fold belt (Krenkel, 1924) that raised by ~500 m since the Pliocene.
 -unclear, rephrase | Rephrased with examples from Fig. 3b:

 We added an explanation and spatial reference, see next column. | The low relief of the southern Galilee basins (*i.e. alluvium covered valleys and intervening small hills*), divides between two segments of the Mesozoic Syrian arc fold belt (Krenkel, 1924; Fig. 3b). *The remnant Mesozoic Syrian arc fold belt that currently builds the Israeli hilly backbone raised by ~500 m since the Pliocene (Fig. 3b- GL, C, UEF, N, SF).* |
| Stratigraphy of the southern Galilee basins"
 -Replace: sedimentary fill. | Followed. | |
| under continental (lacustrine fluvial) environments with phases
 -conditions | Followed. | "…under continental (lacustrine fluvial) conditions with phases…" |
| "Since the early Miocene and until the present, the relatively high rims of the basins have contributed clastics that accumulated in the basins."
 -references. | Followed. References added at the end of sentence: (Shaliv, 1991; Sandler et al., 2004; Rozenbaum et al., 2016). | |
| These studies, focused on localized structures across the southern Galilee basins have been thoroughly studied, | Revised, commas added to the sentence as well. | These studies, focused on localized structures across the southern Galilee basins, *left* the larger, regional, context unresolved. |

| | | |
|---|---|---|
| leaving the larger, regional, context unresolved.
-***Add comma*** were shown.
-replace by: "left" | | |
| …"fundamental questions regarding the origin and development of the lower Galilee…"
-I do not understand the difference between first two options - how do you define and what's the difference between "continental basin" and "full graben"? | This is mainly a question of basin geometries. Architectures as such are discussed and classified in Allen and Allen (2005). All of these varieties were previously suggested for the study area. We added a clarifying sentence at the beginning of the research questions lines, see next column. | "The current study integrates all the previous results with unpublished data to address fundamental questions regarding the origin and development of the lower Galilee. *It surveys the geometry of the basins to clarify regional structural relationships*: is it a single internal continental basin that accumulated sediments from its surrounding rims....." |
| End of Chapter 3:
-I understand that one of the key questions - or the most important one - is the relationship between S Galilee basins and the Irbid rift zone ...? | A following sentence has been added (next column). | "What is the structural and tectonic association between the southern Galilee basins development and the nearby DSF and Levant continental margin? *More specifically, what is the relationship between the southern Galilee basins and the Irbid rift*?" |
| **4. Dataset and Methodology** | | |
| Chapter number revised to 4. | | |
| It includes 85 multi-channel seismic reflection profiles, 506 boreholes, outcrop data, and previous interpretations
-are they all shown on Fig. 3a?
-based on what data? | 1. Typo, changed to 70 due to what shows on Fig. 3 and in Google Earth supplementary.
2. Rephrased: previous *seismic* interpretation | "It includes *70* multi-channel seismic reflection profiles, 506 boreholes, outcrop data, and previous *seismic* interpretations". |

| | | |
|---|---|---|
| The average penetration depth is 500-1000m
-delete penetration
-imaging depth | Followed. | "The average depth _imaging_ is 500-1000m…" |
| Self-addition of the authors for better understanding of this paragraph. | Verified and revised (see next column). | "Stratigraphic, hydrological, geophysical and outcrop datasets collected in the past across the study area are integrated here into a single database _in a WGS 1984-UTM 36N datum-projection, bridging over gaps in vertical and horizontal resolution, reflector amplitudes, processing methods and datum"_. |
| "Structural maps constructed from interpreted key surfaces include associated faults".
-I guess this is more than obvious, structural maps MUST include faults ... | Followed and rephrased as shown in next column. | "_Data were further used for constructing structural maps of key surfaces_". |
| "Previous geological mapping was used to…"
- "Previously published geological maps" or "Results of previous geological mapping" | Followed. | "Results of previous geological mapping _were_ used to… " |
| "Using a 2000 m/sec velocity for the shallow, near-surface beds (weathered beds), enabled correlation between depth and time domains."
- I don't understand this, please describe how well depth data was correlated with time (TWT) seismic data, LVZ is only part of the story here | Two preceding sentences were added to this paragraph. Modifications are shown in italics in the next column. | _"Two velocity surveys were done in the area (Sarid 1 and Revaya 7 wells; SM3, 7, 9). The synthetic seismogram of Revaya 7 well (Frieslander, 1997; Meiler et al., 2008) enabled a reliable stratigraphic correlation with the seismic data. In addition,_ using a 2000 m/sec velocity for the shallow, near-surface beds (weathered beds), enabled correlation between depth and time domains. The wells are found in the supplementary folder (Google Earth archive folder)." |

| | | |
|---|---|---|
| upon digitization of truncation surfaces from previous studies - what are those surfaces? - RTS, anything else? | Yes, indeed. The Galilee "suffered" from an older truncation period at the end of the Senonian (Santonian-Coniacian). It is not so straight forward to differ between the two since the time missing is an amalgamation of these two periods.
We focus on the Oligocene truncation and relevant calculations and estimations in another publication: Wald et al. under review at the Journal of Geodynamics, Tethys Ocean withdrawal and continental peneplanation – an example from the Galilee, northwestern Arabia.
A map showing the older, Senonian truncation is found at the supplementary data: Top of Mt. Scopus Group (Fig. S9:, supplementary data). | |
| I don't fully understand this, explain how / why truncation surface should be considered as layers, and what exactly does it mean for the model. | Quoting from Fig. S11 caption:
Structural map of the top of the Judea Group surface. *Grey polygons- truncated terranes. In these terrains the exposed geological units are older than the mapped surface*. Contours in meters. | |
| I think that this chapter should be significantly rearranged. I don't think that separate description of sedimentary infill and then structural elements (faults, folds) is a good idea. In order to understand subsidence patterns, one needs to properly understand fault activity etc. Therefore, I'd suggest to select several representative seismic profiles located in key parts of the analyzed basin system and calibrated by key wells and describe them including both | We understand the referee's concern, however describing the fill in each basin is tedious and the units are relevant across the basins. That is why after we describe the fill (relatively short chapter, with a lot of references and figures) we define the bordering faults which leads to the model in the discussion.
Our main base for the model are small-scale basins and their border faults. That is why we separate the presentation of them in the Results chapter. Separation and hierarchy (e.g. major vs secondary faults) allows relatively short chapters that provide the most critical issues in regard with basin subsidence. We put effort in providing short and clear opening sentences at the beginning of each paragraph. That way, readers from abroad could interact with the data if they do not want to dive into minor details. Our presentation of | |

| | | |
|---|---|---|
| thickness variations of particular rock formations as well as structural features such as faults, folds, flexures etc. When I was reading this chapter I was constantly jumping between sections 3.1 and 3.2, and it didn't make understanding on this key part of the paper easy ... | subjects is logic (e.g. geographically oriented, from the big picture to the relevant details). | |
| Self-addition of authors. | We added an explanation. | "…and the structural highs (*i.e. uplifted blocks*)" |
| "The latter is thinning towards  towards…"
 -change to thinning | Followed. | The latter is *thinning* towards H2 |
| "The sedimentary fill culminates upwards up…"
 -I'm not sure what does it mean… | Clarified in two sentences as shown on the right column: | "An important surface culminating the Neogene sedimentary fill *is the top of Bira Fm., depicting* a very *mild* relief. *The* Cover Basalt Fm. locally covers *it and provides* a temporal marker." |
| "Analysis of the entire database indicates that the type section is located along the axis…."
 -Explain why in more details. | An explanation sentence has been added, see right column in italics. | Analysis of the entire database indicates that the type section is located along the axis of the Southern Galilee basins (B2, B4, B6, and B7). *Basin depocenters align along a northwest axis (Figs. 1, 7).* Further details from B3, B5, B8-B9 basins complete the section. Additional information from B1, B10, and B11 is provided in the discussion (*location: Fig. 3*). |
| Figure 4 caption: "Contours of equal time gap in million years". | Explanation added to figure caption in section (b). See next column in italics: | (b) Spatial variation in truncation across the Galilee *is a product of kriging interpolation*, *further* represented by |

| | | contours of equal time gap in million years. Black dots mark locations where the youngest unit below RTS and older unit from above are available for quantifying the time gap (*the time gap is discussed in Wald et al., under review*). |
|---|---|---|
| **5.1. Basin fill** | | |
| This description is very detailed, using numerous lithostratigraphic formations and basins / sub-basins. This certainly is correct approach to a PhD thesis aimed mostly at local readers that are familiar with details of local geology but in case of international readers, most probably interested in more general approach towards geological evolution of this area and more generic considerations etc. this level f details might not be necessary and in fact might make proper understanding of main conclusions of this paper difficult. As indicated before, I'd suggest to merge sections on basin fill and faults, and also simplify a bit description focusing on main features like zones of local thickness variations, kinematics of faults | We agree to some extent with the reviewer's approach. Accordingly, local names of structures were replaced by sequential letters and numbers (e.g. B1, B2…B11; H1-H4). Our study area extends next to the seismically active Dead Sea fault plate boundary. The seismic activity along this fault poses a risk to the nearby areas. For example, the largest earthquake of the past decades shook the area in 1984, with a magnitude of 5.4. Its epicenter was in the middle of the study area and not along the Dead Sea fault. Yet, since the subsurface structure of the lower Galilee is not known in detail, the correct earthquake location and hence a reliable risk assessment is lacking. For these reasons we chose the most important feature names to be address in the text, because of their role in the structural architecture of the lower Galilee. | |

| | | |
|---|---|---|
| etc. I'm sure that readers from abroad, without detailed knowledge of geology of Israel, would greatly benefit from this modification. | | |
| Self-correction |  / overlie. | "The Lower Basalt Fm. directly _overlies_ the basin floor". |
| **Figure 5** Seismic profiles are very strongly exaggerated and this makes proper understanding of fault kinematics etc. a bit difficult. I'd suggest to use the entire page in landscape orientation and to stretch these profiles as much as possible in order to make geology looking more realistic | Suggestion taken. Fig. 5 has been extended by x1.5 (X) and squeezed by x0.8 (Y) to nicely fit all sections a-b-c in one page. Reduced vertical exaggeration was applied for figures 8 and 11 as well. The vertical exaggeration is currently stated at the end of each seismic profile figure caption in the manuscript. | |
| **Figure 5** explain why some wells are shown as solid and some as dashed lines | Followed, explained in the caption- a projected well. | …Gideon 1 and G5- Gideon 5, _projected by 1 km from south_ (location: Fig. 3b, GE). |
| **Figure 5** caption: Underlying the commented phrases, corrections shown in italics.

Referee suggestions Flattened ^on the celeste horizon;
*(RTS)
**Unclear, rephrase | Original. Underlying phrases to be corrected (right column).
(c) Same profile, ^flattened relative to the celeste horizon* to image the truncation. The flattening tool enables a comparison between **predating and postdating processes: Cretaceous folding and Neogene subsidence of basins, respectively. Bi-Bira Formation; CS-Clay Series; LwB-Lower Basalt Formation; Groups: Avedat Group; Mt. Scopus Group; Judea Group.

We revised as seen in next column, We added the vertical exaggeration. | (c) Same profile, _flattening of_ the celeste horizon (_RTS_) to image the truncation. The flattening tool enables a comparison between predating and postdating _sedimentary stacks. Flattening the RTS in the seismic software hints at RTS predating and postdating main processes. For example, in Kefar Baruch basin, Cretaceous folding shown by a_ |

| | | |
|---|---|---|
| | | *syncline- predates the RTS, while Neogene subsidence, shown by an accumulation of Neogene sediments- postdates the RTS.* Bi-Bira Formation; CS-Clay Series; LwB-Lower Basalt Formation; Groups: Avedat Group; Mt. Scopus Group; Judea Group. *Vertical exaggeration x5.* |
| Figure 6 caption text edited | Figure 6. *Location of* Lower basalt Formation and Hordos Formation . The westernmost outcrop is along the eastern margins of H1 (location: Fig. 3; DEM- Sneh et al., 2000b). | "Reflectors at the base of the formation onlap an unconformity *(Figs. 5, 8, Figs. S2, S4)*. |
| "Reflectors at the base of the formation onlap an unconformity". - show by arrows on appropriate figure | Followed: Brackets with references to the figures were added. Pointing arrows to the onlap of the Bira Fm. are now seen at Figs. 5 and 8 and also in supplementary figures: Fig. S2, Fig. S4. | |
| "In places it is ..." -overlain | Followed. | |
| Figure 8: -strongly exaggerated, change figure orientation to landscape and stretch this profile | Followed, vertical exaggeration has been reduced from x5 to x2.5. The corrections were added to the end of the figure caption. | "Celeste horizon- RTS.  Orange box- projected location of the Kishon 1 well. Arrows depict onlap of the Bira Fm on the Clay Series. Vertical exaggeration: x2.5." |
| Figure 8: -project this well on seismic profile; | Followed, the well has been projected and it is shown by an orange  dashed rectangle on the seismic section. | "Celeste horizon- RTS. *Orange dashed rectangle- projected location of the Kishon 1 well.* Arrows depict onlap of the Bira Fm on the Clay Series. Vertical exaggeration: x2.5." |
| Figure 8: Two faults were commented the same: (Gevat and Hayogev): | Gevat fault: It is a strike-slip fault with a vertical component. Hayogev fault:This segment of the profile runs parallel to a major structure (Hayogev-Mizra Horst) and is located in a structural junction (see also Fig, 3 and the Google Earth (GE) | |

| | | |
|---|---|---|
| -there are neither thickness variations nor vertical displacements associated with this fault (fault zone), explain how it was interpreted / detected / constrained. | Supplementary data). Due to these facts imaging in this resolution and strike is difficult. The fault is assessed by intervening seismic data. | |
| Figure 8 caption: Self-correction. | To clarify the above comments, a sentence has been added to note the RTS horizon and the Bira Fm. onlap arrows. RTS horizon is shown in celeste. The the vertical exaggeration is noted. | *"Celeste horizon- RTS. Orange dashed rectangle- projected location of the Kishon 1 well. Arrows depict onlap of the Bira Fm on the Clay Series. Vertical exaggeration: x2.5."* |
| **5.2.2 Secondary faults (previously 3.2.2)** | | |
| "(2) faults dividing between basins…" -Dividing what? | dividing between basins, meaning- the faults provide the vertical boundaries of the basins. | |
| "The trace of Yoqneam fault diminishes to the SE until it intersects with Gideon and Hayogev faults in Megiddo region (western margin of B4-5; Fig. 8)". - are all these names of faults necessary? | Yes, we think that this are dominant faults in our model. All of them are shown in the manuscript figures. Also the seismic profiles include them. | |
| Figure 9, middle fault: - structural-thickness interpretation along this fault requires detailed explanation, I do not understand what is going on with dark blue and orange units, especially as there is no displacement on lower seismic horizons. | The figure has been slightly edited to better image the deeper part. This is a combined, long profile and due to this fact the displacements are not clear enough near the marked commented faults. The orange unit is the Hordos Formation (Fig. 2), which is a conglomerate detectable in subsurface seismic data only on the eastern area, near the Dead Sea fault. It predates the basalts and is coeval to them by interfingering. Please see also figures from the Supplementary Material of this paper: Fig. S5, S8, S14. These files also shed light on the | |

| | deeper structure i.e. fault displacement of the Cretaceous-Tertiary units. | |
|---|---|---|
| Figure 9, H4/B7 faults:
- how these faults were constrained ...? | A comment regarding the deeper section interpretation was already part of the original caption. These faults were constrained by high resolution seismic profiles shown in Fig. 3 at the Bet Shean basin (B7). Please also look at Fig. S5, S8, S14 (Supplementary Material). | |
| Figure 9 caption
: "Cretaceous units at the syncline were interpreted…"
-self revision to clarify | Clarifications added to figure caption, to meet the referee's comments for this figure.
A comment regarding the deeper section interpretation was added to the caption and also "No vertical exaggeration". | "Cretaceous units at the syncline (*Kefar Baruch, B2) and in the eastern B7 area* were interpreted…"
"*No vertical exaggeration*" |
| **6. Discussion** | | |
| | | |
| -Very strongly exaggerated | This section has been re-edited to reduce the exaggeration:
x1.5 stretch for the horizontal scale
x0.8 squeeze for the vertical scale.
Figure 11 caption revised in italics & underlined text. | "Multi-channel seismic reflection profile line MI-2178. *Strike-slip faults with a normal component in the frame of Plio-Pleistocene* lateral shear adjacent to Mt. Carmel and Tivon blocks. Normal faulting vertically offsets basin fill units. Post-Bira Formation folding (postdating uppermost Miocene-Pliocene) *is assigned to the strike-slip shear on the originally normal faults.* A paleo alluvial fan, predating the vertical offset, is depicted by the Clay Series. *Celeste horizon- RTS;* dashed lines-projected wells. NHL- Nahalal; KY- Kefar Yehoshua. Location: Figure 3; Unit color code- Figure 2. *Vertical exaggeration: x3.* |
| Kefar Baruch, west fault branch
- what's the kinematics of this fault? | Strike-slip with normal component. The caption has been corrected to include an explanation for fault types. | |

| | | |
|---|---|---|
| Kefar Baruch fault
- normal faulting was suggested but there is significant displacement at top Lower Basal Formation and top Clay Series and no displacement at deeper seismic horizons, how this could be explained? | Corrected. Strike-slip with normal component. The caption has been corrected to include an explanation for fault types.
See also the remark to Nahalal fault.
These faults have a lateral displacement component as well and they are part of the Carmel-Gilboa fault system off the Dead Sea fault. EQ focal solutions back up our interpretations. | |
| Gevat fault
- normal faulting was suggested but there is significant displacement at top Lower Basal Formation and top Clay Series and no displacement at deeper seismic horizons, how this could be explained? | Corrected. Strike-slip with normal component. The caption has been corrected to include an explanation for fault types. See also the remark to Nahalal fault. These faults have a lateral displacement component as well and they are part of the Carmel-Gilboa fault system off the Dead Sea fault. EQ focal solutions back up our interpretations. | |
| Nahalal fault
- normal faulting was suggested but no normal displacement at top of Mt Scopus Group and top of Judea Group is observed, how this could be explained? | The figure caption explains that the faults imaged are strike-slip with normal component. **The caption has been rephrased to further verify the fault types**.
The Nahalal fault is a strike-slip fault with a very distinct flower structure also verified by cross lines and from EQ focal solution.
We focus on the faulting in a future publication. This fault probably began as normal, but the later deformation obscures the reflectors in the fault plane and fault branches in addition to low resolution above 0.5 msec two way time.
Kefar Baruch and Gevat faults are normal but have a strike-slip sense of movement that might explain the folding of Neogene layers adjacent to the fault planes. The graben between these two faults is verified in potential methods | Figure 11. Multi-channel seismic reflection profile line MI-2178. *Strike-slip faults with a normal component in the frame of Plio-Pleistocene* lateral shear adjacent to Mt. Carmel and Tivon blocks. Normal faulting vertically offsets basin fill units. Post-Bira Formation folding (postdating uppermost Miocene-Pliocene) *is assigned to the strike-slip shear on the originally normal faults*. A paleo alluvial fan, predating the vertical offset, is depicted by the Clay Series. *Celeste horizon- RTS; dashed lines- projected wells.* NHL- Nahalal; KY- Kefar Yehoshua. Location: Figure 3; Unit color code- Figure 2. *Vertical exaggeration: x3.* |

| | (Segev and Rybakov, 2011) and by crossing seismic lines interpreted herein. | |
|---|---|---|
| **6.1.1. First stage (20-9 Ma)** | | |
| General remark:
-simple 2D model showing main subsidence centers and key fault zones would greatly help to understand proposed model | Figure 12 is our suggested model in map view. In addition, Fig. 1B nicely shows the subsidence centers along the Irbid rift. | |
| "However, remains of Hordos Fm. are not restricted the subsiding basins"
-unclear rephrase | Rephrased. | "However, remains of Hordos Fm. are not restricted the subsiding basins. *Their extent is larger than the current northwest array of basins.* They appear in sporadic outcrops..." |
| "During the mid-Miocene, normal displacements along faults facilitated deepening of the basins".

-I'm having problems with understanding of the proposed normal faulting shown on interpreted seismic profiles (e.g. Fig. 11), please provide proper and comprehensive explanation for this. | Due to post-Miocene shear the fault planes no longer show pure normal faulting but rather amalgamate strike-slip faulting as well. This issues will be discussed in a future paper. Figure 11 shows a seismic profile collected very close to a major structural junction between two sets of faults and it shows strike-slip deformation (EQ focal solution works). That is why we mentioned figures 7, 10c and 12.
Most of the Neogene sedimentary stacks clearly show the downfaulted blocks. The isopach map of the Lower Basalt shows it nicely. The problem you mention is with the Cretaceous portion of the seismic sections. The Cretaceous units are folded and then faulted. Not always does the resolution of this continental seismic reflection data enable to clearly image the fault planes at this depth. | |
| Figure 12:
-what are dotted lines? | Added to figure caption: Dashed lines portray the structural highs (H1-4 in Fig. 3a). | "(a) Structural map of the top Avedat Group. *Dashed lines portray the structural highs (H1-4 in Fig. 3a).* |
| **6.1.2  Second stage (9-5 Ma)** | | |
| General remark chapter 6.1.2 | Thank you for this comment. However, the names were chosen carefully, always with spatial reference (location in | |

| | | |
|---|---|---|
| -text below is full of names of local rock formations, local faults etc., it is not easy to follow this, try to simplify this description if possible | Google Earth, map). Localities were chosen to verify and discuss major ideas concerning the second stage of basin subsidence. These details are important since this period encompasses various regional phenomena including the MSC (Messinian Salt Crisis) followed by the flooding of the valleys … In addition a new faulting mechanism became dominant (obeying the Dead Sea stress field). Examples here are important to strengthen our model. | |
| "record does not contradict additional marine intercalations between 5.4-5.25 Ma (e.g., Haq et al., 1987; Müller and Hsu, 1987). This deduction is also supported by mega-fauna (Shaliv, 1991), ostrea lumashell unusual facies in outcrops of southern B9 (Schulman, 1962) and lithological resemblance of the latter evidence and those of the southern Galilee basins (Michelson and Lipson-Benitah, 1986)".
- clarify why this should be regarded as support | We added "marine" for verification (underlined and in italics in next column). Also marine succession at the end of sentence. | "This deduction is also supported by _marine_ mega-fauna (Shaliv, 1991), ostrea lumashell unusual facies in outcrops of southern B9 (Schulman, 1962) and lithological resemblance of the latter evidence and those of the southern Galilee basins _marine succession_ (Michelson and Lipson-Benitah, 1986)". |
| "Further east the facies shifts land locked lake environments of the Sedom Fm.."

-Unclear, rephrase. | The paragraph has been rephrased to better explain the facies shift of the Bira Fm. from "Pattish" to "Bira" to "Sedom" following Gvirtzman et al. (2011).
See new text in italics and underlined in next column. | "Gvirtzman et al. (2011) describe a lateral facial shift during the late Miocene (Fig. 13 in Gvirtzman et al., 2011): Pattish Fm. _represents the first facies_ of a continental shelf (i.e., marine) environment. The transition to the _second_ lacustrine |

| | | floodplain facies of Bira Fm. is located on the eastern flank of H1, next to the intersection between Sede Yaakov and western Gilboa faults in Tel Kashish (Figs. 3, 10c, Fig. S12; Zilberman and Sandler, 2013). Further east, the *third facies of the Bira Fm.* is represented by the land locked lake environments of the Sedom Fm (*i.e. along the Dead Sea fault*)". |
|---|---|---|
| **6.2 Tectonic classification of the basins during the two stages** | | |
| "The Galilee basins developed during the Neogene  two major structural processes -replace by due to | Followed. | "The Galilee basins developed during the Neogene *due to* two major structural processes". |
| "…during the first stage…" -when exactly? | | "…during the first stage *(20-9 Ma)*" |
| "The thinning of the Lower Basalt Fm. to the northwest (Figs. 6, 10) [1]points to shallowing of the basin floor in that direction, and hence to a reduction in regional extension towards the continental margin in the west. At this stage, the structure of the basins and their [2]dimensions are equivalent to the definition of intraplate basins that form during rifting (Evison, 1959; Bosworth, 1994; Busby and Ingersoll, 1995; Allen and Allen, 2005; Morley et al., 2004) as well as | Paragraph revised: See next column in italics/underlined. Point by point numbered answers here. 1- That is true. We fixed and elaborated. See next column. Volcanic sources diminish to this direction, away from the plume channeled from Afar, along the western side of Arabia and in DSF vicinity. 2- Deleted to: "At this stage, the structure of the basins and their dimensions are equivalent to intraplate basins". 3- Remark taken. Rephrased. | "The thinning of the Lower Basalt Fm. to the northwest (Figs. 6, 10) *supports diminish of volcanic sources as well as* shallowing of the basin floor in that direction. *The Lower Basalt does not cross H1 (Fig. 3) to the west. This trend suggests a* reduction in regional extension towards the continental margin in the west, *previously assessed by Freund (1970)*. At this stage, the structure of the basins and their dimensions are equivalent to intraplate *grabens and half-graben* basins that form during *intra-continental* rifting (Evison, 1959; Bosworth, 1994; Busby and Ingersoll, 1995; Allen and Allen, 2005; Morley et al., 2004). Previous studies showed the development of the *Irbid (also* |

| | | |
|---|---|---|
| to the [3]grabens and half-grabens of intra-continental rifts (Bosworth, 1994).

1- Not necessarily, basaltic covers might be characterized by lateral thickness variations solely due to different distances from the source area.
2- Dimension could not be equivalent to definition.
3-What sort of difference there is between "intraplate basins formed during rifting" and grabens and "half-grabens of intra-continental rifts"? | | *referred to as Qishon-Sirhan or Azraq-Sirhan)* rift during the Oligocene-Miocene in a northwesterly direction (Shaliv, 1991; Schattner et al., 2006a; Segev et al., 2014)". |
| "Previous studies showed the development of the **[1]***Qishon-Sirhan rift* during the Oligocene-Miocene in a northwesterly direction (Shaliv, 1991; Schattner et al., 2006a; Segev et al., 2014). **[2]**Results of the present study claim this rift comprises the southern Galilee Basins during their first stage of development.
1-what is this rift, where located, and how related to the entire story described in this paper? | 1- Revised, next column
2-This sentence was deleted | "Previous studies showed the development of the *Irbid* rift (*also referred to as Qishon-Sirhan or Azraq-Sirhan*) during the Oligocene-Miocene in a northwesterly direction..." |

| | | |
|---|---|---|
| 2-this is very unclear statement | | |
| Basins [1]subside vertically and [2]extend perpendicularly to 5 the principal axis of the first stage basins, [3]while structural highs separate between them (Fig. 12).
1- subsided
2-extended
3-unclear statement | 1,2-followed.
3-Explanations added, revised as seen in next column. | "Basins subsided vertically and extended perpendicularly to the principal axis of the first stage basins, while *uplifted blocks* (*i.e. structural highs*) separate between them *in a NNE direction (Fig. 12).*
This is further explained in the caption of figure 12 now (dashed lines in 12a)". |
| The highs are accommodation zones, structurally equivalent to the separators between basins along the East African Rift.
-? | Rephrased to: intervening block separators | The highs are accommodation zones, structurally equivalent to *the intervening block separators* between basins along the East African Rift. |
| "These studies also show that basins…"
- which studies - referred to in the previous sentence? | Yes, exactly. Rephrased to: The latter studies... | *"The latter studies* also show that basins…" |
| "...some of these rifts may succeed and continue to drift, while others fail".

-rifts could not drift apart, plates separated by a rift zone can | "drift" changed to "open" | "…some of these rifts may succeed and continue to *open*, while others fail". |
| "The two stages recorded here occurred alongside the initiation of motion along the nearby DSF plate boundary. While motion took place along | Yes. We deleted and then thoroughly revised the last two sentences into a clearer paragraph regarding the interaction between rift and the DSF (see next column).
We added a reference to Fig. 1b. | "Interaction between the Dead Sea fault and the Irbid rift is depicted by the deep depocenter of Bet Shean basin (B7) at the then junction area (Fig. 7; pre-lateral displacement on the Dead Sea fault). Volcanism initiation is also suggested as 17 Ma for the Galilee (Rozenbaum et al., 2016; Shaliv et al., 1991). Transform-rift |

| | | |
|---|---|---|
| the entire boundary between the Arabian and Sinai plates, intracontinental basins subsided only across the southern Galilee".

 - unclear, rephrase and provide more comprehensive explanation | | interaction adjacent to continental margin is manifested by NW-striking faults within the Galilee and NE-striking faults within the Golan Heights (location: Fig. 3). This process signifies the crossing of the Irbid rift into the other side of the DSF (Fig. 1b; Segev et al., 2014). Our study supports the numerical modelling of Segev et al. (2014) by showing that the active rifting of the Irbid rift on the western side of the DSF succeeded in opening basins by cutting across the Levant continental margin (Fig. 1b)". |
| **6.3 Structural stress field transitions along the plate boundary** | | |
| - transitions of what? | Title corrected: Structural stress field transitions | |
| Are buried  the subsurface of the southern Galilee basins -replace with in | Followed. | |
| "The Red Sea extensional regime..." -orientiation?" | brackets added: (N60°E extension) | " The Red Sea extensional regime *(N60°E extension)* …" |
| "The ENE axis extension (…)" - ? | Rephrased to: "The N60°E extension (McClay..." | " The *N60°E* extension…" |
| "The NW trending faults developed across the study area are part of a larger set of the western Arabian plate. - faults are set of the plate ...? I do not understand this | This was a typo. Rephrased, Najd fault system has been included in the revised Fig. 1. | "The NW trending faults developed across the study area are part of *larger fault systems extending across the western Arabian plate (Fig. 1)*". |
| -quite a lot of statements from this chapter seem not to be based on results of seismic data interpretation presented in this paper, it is not very easy | This chapter begins with vast evidence from the regional geology and 'channels' the reader into the Galilee study area. We find it important to relate to the regional tectonics since the Red Sea and the Irbid rift are coeval in their evolution and both interact with the embrionic Dead Sea fault. | |

| | | |
|---|---|---|
| to understand how this is all interrelated ... | | |
| "…resulted in tectonic quiescence in Suez (Bayer et al., 1989)..."
-Red Sea rift | OK. Bayer showed evidence from Suez the equivalent rift on the Sinai sub-plate (west of the plate boundary.
Revised – see next column. | "... resulted in tectonic quiescence in *the Suez portion of the Red Sea rift*..." |
| " In between the two rift systems, the Negev ceased to subside (Zilberman and Calvo, 2013; location- Fig. 1);"
-what do you mean by this? | Regions between the two rifts of Suez and Galilee. brackets added
(i.e. Suez and Galilee)
Negev (southern Israel) | " In between the two rift systems (*i.e. Suez and Galilee*), the Negev (*southern Israel*) ceased to subside (Zilberman and Calvo, 2013; location- Fig. 1);" |
| "During the same time window, a [1]numerical simulation shows a depression that subsided along the [2]Sirhan trajectory, still not entirely affected by the displacement along the intersecting DSF".
1-nothing of this kind has been described in this paper, give more details and references.
2-how do you define this "trajectory"? | Revised and answered, with references to the numerical model already given in the manuscript.
**1-**This important work of numerical simulations was referred to in the Introduction and in Chapter 3.2.2:
"A series of NNE to NE-trending normal faults divide between the basins and structural highs of the southern Galilee. The faults are nearly perpendicular to the axis of the basins complex. Seismic data show that displacements across these faults are mainly vertical with a horizontal component. Regional numerical modelling of Lyakhovsky et al (2012) followed by a review of rift-transform interaction adjacent to continental margins (Segev et al., 2014), has predicted rift-perpendicular features. Locally, these faults, structural highs, and basins between them are evident from the structural map of top Avedat Gr. that consist the floor of most of the basins (Fig. 12). The following paragraphs describe the division along the major axis, from NW to SE".

**We also refer and discuss this at the end of the preceding chapter, 6.3 (previously 4.3):
"Interaction between the Dead Sea fault and the Irbid rift is depicted by the deep depocenter of Bet Shean basin (B7) at | 1-Refereces given in previous column.
2-"...shows a depression that subsided along the Irbid rift NW-trending axis, still not entirely affected ..." |

| | | |
|---|---|---|
| | the then junction area (Fig. 7; pre-lateral displacement on the Dead Sea fault). Volcanism initiation is also suggested as 17 Ma for the Galilee (Rozenbaum et al., 2016; Shaliv et al., 1991). Transform-rift interaction adjacent to continental margin is manifested by NW-striking faults within the Galilee and NE-striking faults within the Golan Heights (location: Fig. 3). This process signifies the crossing of the Irbid rift into the other side of the DSF (Segev et al., 2014). Our study supports the numerical modelling of Segev et al. (2014) by showing that the active rifting of the Irbid rift on the western side of the DSF succeeded in opening basins by cutting across the Levant continental margin".

\*\* Finally, we refer to these works at the end of the Conclusions chapter.

**2**-Trajectory is an axis of motion, taken from physics. We decided to revise, see (2) next column. | |
| "During that time, volcanic activity stopped in Syria (Mouty et al., 1992)."
-why this is important, and what's the conclusion of this statement? | We decided to erase this sentence along with the reference, it is irrelevant to this chapter. | |
| "…Galilee basins continued to extend during the upper Miocene.
-late Miocene | Followed. | |
| "Freund (1970) calculates…"
-calculated | Followed. | |
| "… and 7% in B7 near the DSF".
-with all those symbols and abbreviations, it is difficult to | The B series abbreviations has been fixed to reduce the use of full basin names each time in text. However, following your comment where possible we add an explanatory remark or brackets. | |

| | | |
|---|---|---|
| follow this paper ... Try to simplify this as much as possible | | |
| Self-addition: a new paragraph added in reply to referee # 1 comment; not included in the pdf submitted by referee #2 with comments. We quote the whole new paragraph added just before figure 13 in our revised manuscript. | "Plio-Pleistocene geodynamic analysis poses the study area as a seismogenic branch off the DSF plate boundary. The Primary Deformation Zone (PDZ) is expressed by a northwest oriented cross-cutting shear that overcomes basin subsidence. Earthquake epicenter distribution and mechanisms, GPS measurements and regional studies point to a seismogenic zone located at 9-17 kilometres beneath the surface (Eyal and Reches, 1983; Ron and Eyal, 1985; Ben-Avraham and Ginzburg, 1990; Eyal, 1996; Hofstetter et al., 1996; Hardy et al., 2010; Salamon et al., 2006; Gomez et al., 2007; Shamir, 2007; Marco, 2007; Sadeh et al., 2012; Palano et al., 2013). Our tectonic analysis of the Galilean sheared margins in the frame of the Dead Sea fault localization process will be published in a separate paper (Wald, 2016)". | |
| Figure 13 caption
-this should be described in the text, also because McClay & Bonora 2001 paper is about restraining stepovers not related to any focused subsidence (i.e. releasing stepovers) | Indeed. Caption edited. Especially (b).
Bonora and McClay do research step-over, restraining jogs. The revised text deals with this issue in the last paragraph of chapter 6.3, preceding Figure 13.
The caption has been edited as follows (italics & underlined in next column).

We will focus on post-Miocene shear in a future paper. The main idea is:
Complex Pliocene-Pleistocene wrench shear dissects the Galilee basins in a NW direction. The PDZ connects the faulted northern faces of Mt. Gilboa and Mt. Carmel. Strain direction parallels the central axis of the basins (WNW oriented). Post-Miocene faulting patterns locally cover the original architecture of the basins. | Figure 13. (a) *Current* plan view of the northwest trending Irbid rift dissected by the Dead Sea fault plate boundary. *(b) Neogene basin subsidence across the Galilee during the Irbid rifting (marked as 1st stage).* NNE elongation provoked extension across the interpreted normal faults (marked by celeste lines). The 2nd stage reflects the Dead Sea Fault (DSF) stress regime, *during which subsidence, normal faulting and graben formation decrease while complex strike-slip faulting characterizes the strain style. An establishment of a left-lateral strike-slip Primary Deformation Zone- PDZ, modified after McClay and Bonora, 2001*. |

| | | |
|---|---|---|
| Figure 13c
- why dotted line was used here? | A sentence was added to the end of the figure caption:
Dotted lines: less verified fault planes. | |
| -why both normal and strike-slip kinematics is shown for this fault? | This is a normal fault that has been reactivated as a strike-slip. Very common in the Plio-Pleistocene deformation of the area. | |
| "With enhancement of motion along the DSF during the lower Pliocene around 5 Ma…"
-Early instead of lower Pliocene | Followed. | |
| **7. Conclusions** | | |
| "The Galilee basins subsided along the northwestern front of the Sirhan rift".

-Irbid? - you have to be consisted with terminology for the entire paper | Followed, corrected to Irbid. Throughout the manuscript. | |
| [1]Structural highs that divide between the first-stage basins [2]remain high during the second stage
1-unclear
2-remained | Followed:
1- "Structural highs *(i.e. blocks)* that divide between the first-stage basins highs…"

2- "…*remained* high during the second stage" | 1-"Structural highs *(i.e. blocks)* that divide between the first-stage basins highs…" |
| "The *general shear* distorts the original structure of the first stage…"
-what exactly do you mean by this? | This sentence has been revised to clarify (next column, italics & underlined text) | "The *shear* distorts the original structure of the first stage basins north and south of the major NW-trending axis, *in a manner that today these periphery early Neogene basins have been uplifted and weathered*". |

| | | |
|---|---|---|
| "…. Their shape and arrangement were constrained by two main rheological features – [1]the bounds of a releasing jog along the PDZ and the [2]acquaintance with a more cohesive crust at the peripheral area, perhaps a "locked zone" (see Lyakhovsky et al., 2012; Segev et al., 2014). However, [3]neither of these seems to have caused the cessation of rifting".
 1-i.e. ...?
 2-Explain in more details.
 3- why? | **1**- (i.e. Carmel-Gilboa fault line)
 **2**- A sentence has been added to clarify about the "locked zone" concept in rift propagation.
 References were added to support:
 Courtillot, V. Armijo, R. and Tapponnier, P., 1987. Kinematics of the Sinai triple junction and a two-phase model of Arabia-Africa rifting DOI: 10.1016/0040-1951(87)90184-3, Tectonophysics, 141(1-3), 181-190.

 Dunbar, J.A. and Sawyer, D.S., 1996. Three-dimensional dynamical model of continental rift propagation and margin plateau formation. Journal of Geophysical Research, 101(B12), 27,845- 27,863.

 3-Edited:
 However, *following the numerical modeling results, neither of these seems to have caused the cessation*...
 This is due to the results of numerical models of the preceding sentence (see references). | Their shape and arrangement were constrained by two main rheological features – 1the bounds of a releasing jog along the PDZ *(i.e. Carmel-Gilboa fault line)* and the acquaintance with a more cohesive crust at the peripheral area, perhaps a "locked zone" (see Lyakhovsky et al., 2012; Segev et al., 2014). *Locked zones involve pre-existing discontinuities suchlike transition between oceanic and continental crust types or perpendicular faulting arrays (Courtllot et al., 1987; Dunbar and Sawyer, 1996). H*owever, *following the numerical modelling results,* neither of these seems to have caused the cessation of rifting". |
| "Based on this case study we suggest that the rift did not fail but rather faded and was taken over by a more dominant stress regime."
 - but not the entire Idris rift, just its NW tip in the present-day Israel, correct? If so, add few words how this structural story might be manifested in the main rift zone. | Thanks for this request.
 We clarified this issue by adding the word "front" in these two sentences:

 ""Based on this case study we suggest that the *rift front* did not fail but rather faded and was taken over by a more dominant stress regime. Otherwise, basins of this failing rift *front* could have simply died out".

 We do not want to confuse the readers with the eastern portion of the rift, that had been cut off and uplifted due to the lateral relative plate motion along the Dead Sea fault. | ""Based on this case study we suggest that the *rift front* did not fail but rather faded and was taken over by a more dominant stress regime. Otherwise, basins of this failing rift *front* could have simply died out". |

| | | |
|---|---|---|
| Acknowledgement: We thank the Kingdom Suite and Schlumberger-Petrel for providing academic licenses that facilitated this study

-are there any results from Petrel shown in this paper? | Yes. Figure 13 is a slice from the geological model built in Petrel. The two stages of subsidence have been calculated in Petrel. Structural maps (Figs. 7,12a; Figs S9, S11 in the supplementary) have been built from surfaces exported from the Kingdom Suite software to Petrel. | |

[revised manuscript text omitted]

diminished

| Page 11: [1] Deleted | Reli | 11/24/2018 3:59:00 PM |

diminished

| Page 11: [1] Deleted | Reli | 11/24/2018 3:59:00 PM |

diminished

| Page 11: [1] Deleted | Reli | 11/24/2018 3:59:00 PM |

diminished

| Page 11: [1] Deleted | Reli | 11/24/2018 3:59:00 PM |

diminished

| Page 11: [1] Deleted | Reli | 11/24/2018 3:59:00 PM |

diminished

| Page 11: [1] Deleted | Reli | 11/24/2018 3:59:00 PM |

diminished

| Page 11: [1] Deleted | Reli | 11/24/2018 3:59:00 PM |

diminished

| Page 11: [1] Deleted | Reli | 11/24/2018 3:59:00 PM |

diminished

| Page 11: [2] Deleted | Reli | 11/24/2018 7:34:00 PM |

[Figure]

Page 23: [3] Deleted                                **Reli**                                11/29/2018 1:42:00 PM